# Federated Node-Level Clustering Network with Cross-Subgraph Link Mending

**Jingxin Liu**[1]   **Renda Han**[2]   **Wenxuan Tu**[2]   **Haotian Wang**[2]   **Junlong Wu**[2]   **Jieren Cheng**[2]

## Abstract

Subgraphs of a complete graph are usually distributed across multiple devices and can only be accessed locally because the raw data cannot be directly shared. However, existing node-level federated graph learning suffers from at least one of the following issues: 1) heavily relying on labeled graph samples that are difficult to obtain in real-world applications, and 2) partitioning a complete graph into several subgraphs inevitably causes missing links, leading to sub-optimal sample representations. To solve these issues, we propose a novel Federated Node-level Clustering Network (FedNCN), which mends the destroyed cross-subgraph links using clustering prior knowledge. Specifically, within each client, we first design an MLP-based projector to implicitly preserve key clustering properties of a subgraph in a denoising learning-like manner, and then upload the resultant clustering signals that are hard to reconstruct for subsequent cross-subgraph links restoration. In the server, we maximize the potential affinity between subgraphs stemming from clustering signals by graph similarity estimation and minimize redundant links via the N-Cut criterion. Moreover, we employ a GNN-based generator to learn consensus prototypes from this mended graph, enabling the MLP-GNN joint-optimized learner to enhance data privacy during data transmission and further promote the local model for better clustering. Extensive experiments demonstrate the superiority of FedNCN.



*Figure 1.* Research motivation for cross-subgraph link mending. We visualize the encoded raw graph structure using heat maps. The similarities of the destroyed and mended topologies relative to the original topology are 44.74% and 78.45%, respectively.

## 1. Introduction

Graph machine learning (GML) is a well-established technique that harnesses the structure of graph data to facilitate various learning tasks. A large amount of GML methods have recently been developed and achieved satisfactory performance (Liang et al., 2023). Commonly, almost all of them assume that the graph data is centralized and available unconditionally. However, in numerous real-world scenarios, such as medical data analysis (Huang et al., 2024), smart transportation (Chen et al., 2024), and social networks (Chen et al., 2022), a complete graph is usually partitioned into multiple subgraphs, with each one distributed to an isolated client without directly exposing the raw data. In this background, node-level federated graph learning (FGL) has emerged as a promising solution and has attracted a lot of research attention in recent years.

To be brief, node-level FGL is a machine learning paradigm composed of multiple isolated clients and an independent server (Baek et al., 2023; Zhu et al., 2024; Li et al., 2024) to group the distributed subgraphs into categories while ensuring private data is not directly shared. Typically, each client trains local model parameters with its own label signals and sends them to the server, while the server integrates the uploaded signals to compute a consensus one that is subsequently sent back to all clients to update the local models. Despite their encouraging successes, we find that these studies have at least one non-negligible limitation: 1) heavily relying on labeled graph samples that are difficult to obtain in real-world applications (Zhang et al., 2021; Tan

---

[1]School of Cyberspace Security, Hainan University, Haikou, China [2]School of Computer Science and Technology, Hainan University, Haikou, China. Correspondence to: Wenxuan Tu <twx@hainanu.edu.cn>, Jieren Cheng <992730@hainanu.edu.cn>.

et al., 2024; Wan et al., 2024). Once reliable supervised signals are unavailable, existing methods struggle to effectively learn high-quality representations, which adversely affects the quality of multi-source information negotiation at the server; and 2) partitioning a complete graph into several subgraphs of random sizes inevitably causes substantial amounts of missing links between clients (Zhu et al., 2024; Baek et al., 2023). As illustrated in Fig. 1, taking the results on Citeseer as an example, more than 55% of the sample connections are destroyed during the graph partitioning process. As the number of clients increases, the disruption of links between samples would become more severe. Consequently, it is urgent to develop a novel unsupervised node-level FGL framework to handle the aforementioned issues.

An intuitive solution involves leveraging clustering signals to facilitate the restoration of missing links across subgraphs on the server, where the mended graph is in turn utilized to boost the performance of each client. To fulfill this, there are two key challenges to be solved: 1) how to collect and preserve key clustering properties from multiple unlabeled subgraphs without directly sharing private information; and 2) how to establish correct topology relationships among subgraphs with the aid of provided pseudo-supervised information. For the first challenge, inspired by the prototype learning (Liu et al., 2025; Wan et al., 2024), we first select representative nodes from each cluster within the client, and then design a clustering projector to generate hard-to-reconstruct counterparts of the selected samples as well as preserve the clustering characteristics from the local model. For the second challenge, we attempt to construct the potential affinity between cross-subgraphs stemming from uploaded clustering signals via graph similarity estimation.

Based on the above observations, we propose a novel federated node-level clustering framework termed FedNCN. The core idea of FedNCN is to mend the destroyed links using clustering prior knowledge. Specifically, within each client, we develop an MLP-based projector to implicitly preserve the key clustering properties (i.e., prototypes) of each subgraph through a denoising-like learning approach. Subsequently, in the server, we maximize the potential affinity of subgraphs derived from uploaded clustering signals that are hard to reconstruct into the raw data via graph similarity estimation. Meanwhile, we employ the improved N-Cut operation to decrease the redundant sample connections within the mended graph as much as possible. After that, a GNN-based generator is designed to learn consensus prototypes based on the mended graph, where the MLP-GNN joint-optimized learner transmits the updated prototypes and model parameters back to each client while protecting data privacy during transmission. As shown in Fig. 1, with the proposed cross-subgraph link mending scheme, our method can effectively restore the missing links from 44.74% to 78.45% for better clustering performance (see Section 4.2).

Our main contributions can be summarized as follows:

- **New research task.** To the best of our knowledge, we make the first attempt to tackle the issue of link missing caused by graph partition in federated node-level clustering.

- **Novel FGL framework.** A Federated Node-level Clustering Network (FedNCN) is proposed. It not only effectively mends cross-subgraph links, but also promotes the great encoding capacity of the local model for better clustering.

- **Better clustering results.** Extensive experiments on five graph benchmark datasets demonstrate the effectiveness and superiority of the proposed FedNCN compared to its competitors.

## 2. Related Work

### 2.1. Federated Graph Learning

Federated Learning (FL) is a widely used distributed machine learning framework that enables multiple clients to collaborate in training a global model through a central server, while avoiding direct sharing of raw data (Wang et al., 2021a;b; 2023a; 2024b;c;e; Meng et al., 2024; Wang et al., 2023b; 2024a;d). Due to its ability to effectively leverage various technologies for application extension, FL has been rapidly developed in recent years, particularly in graph data analysis. For example, FedSage+ (Zhang et al., 2021) utilizes node labels to design a missing neighbor generator, addressing the issue of missing links in distributed subgraph systems. Moreover, FedPUB (Baek et al., 2023) leverages labeled data to generate a global random graph, which is then fed back to clients to obtain multiple functional embeddings. These embeddings are used to compute similarities for identifying different communities, thereby mitigating multi-source heterogeneity. Similarly, FedTAD (Zhu et al., 2024) assesses the reliability of node class knowledge in a topology-aware manner, which is then uploaded to the server to guide pseudo-graph generation for better classification. Previous studies have demonstrated that the FGL framework can be integrated with some advanced techniques and achieve promising performance with the aid of data labels. In contrast, to adapt to the unlabeled scenario, we design a clustering projector that generates hard-to-reconstruct counterparts while preserving the clustering properties of each cluster for better clustering.

### 2.2. Attributed Graph Clustering

Benefiting from the strong generalization ability of the GNN in handling graph data, significant progress has been made in node-level clustering tasks in recent years (Tu et al., 2021; Li et al., 2022; Gong et al., 2022a;b; Pan & Kang, 2023;

Guan et al., 2025). CCGC (Yang et al., 2023) constructs two views of the complete graph and utilizes a siamese encoder to guide the generation of positive and negative sample pairs, thereby improving clustering performance. Similarly, MAGC (Lin et al., 2021) mines complementary information from multi-view data using weight factors to learn consistency and discriminative relationships. Furthermore, AMGC (Tu et al., 2024a) designs a unified framework that alternately optimizes clustering and attribute imputation processes on a single graph. Commonly, these methods assume that the graph data is centralized. However, this assumption is overly strict in real-world scenarios, as subgraphs of a complete graph are usually distributed across multiple devices and are accessible only locally, which inevitably leads to missing links between subgraphs. To this end, we construct cross-subgraph potential affinity relationships based on prior clustering knowledge to restore the destroyed links.

## 3. Methodology

Fig. 2 shows the architecture of the FedNCN. The core idea is to mend the cross-subgraph missing links to enhance the clustering performance of each client in an unlabeled circumstance. This process mainly consists of three stages: local model learning, cross-subgraph link mending, and global knowledge sharing.

### 3.1. Notations

Denote by $\mathcal{G} = \{\mathcal{V}, \mathcal{E}\}$ a complete undirected graph with $O$ cluster centers, where $\mathcal{V}$ and $\mathcal{E}$ are the sets of nodes and edges, respectively. Suppose the $\mathcal{G}$ is divided into $M$ subgraphs, which are assigned to $M$ clients. For simplicity, we take the local learning over a single subgraph $\bar{\mathcal{G}}$ as an example and consider other graphs similarly. Suppose that $\bar{\mathcal{G}}$ contains $\bar{O}$ cluster centers and $\bar{N}$ nodes. Let $\bar{\mathbf{X}} \in \mathbb{R}^{\bar{N} \times d}$ and $\bar{\mathbf{A}} \in \mathbb{R}^{\bar{N} \times \bar{N}}$ denote the attribute and original adjacency matrices, where $d$ is the attribute dimension. The corresponding degree matrix is $\bar{\mathbf{D}} \in \mathbb{R}^{\bar{N} \times \bar{N}}$. With $\bar{\mathbf{D}}$, the $\bar{\mathbf{A}}$ is further normalized as $\widetilde{\mathbf{A}} \in \mathbb{R}^{\bar{N} \times \bar{N}}$ by calculating $\bar{\mathbf{D}}^{-\frac{1}{2}}(\bar{\mathbf{A}} + \mathbf{I})\bar{\mathbf{D}}^{-\frac{1}{2}}$, where $\mathbf{I} \in \mathbb{R}^{\bar{N} \times \bar{N}}$ indicates identity matrix. All used notations are summarized in Appendix A.

### 3.2. Local Model Learning

This section introduces the local model learning for a single subgraph, and the goal is to preserve the key information of each cluster within each unlabeled subgraph. We first employ a GNN model to capture the node embeddings of the subgraph as follows:

$$\mathbf{Z}^{(l)} = \sigma(\widetilde{\mathbf{A}}\mathbf{Z}^{(l-1)}\bar{\mathbf{W}}^{(l)}), \qquad (1)$$

where $\mathbf{Z}^{(l)} \in \mathbb{R}^{\bar{N} \times d'}$ denotes the node embeddings of the $l$-th encoder layer, and $\mathbf{Z}^{(0)}$ equals to $\bar{\mathbf{X}}$. $\bar{\mathbf{W}}^{(l)}$ is the learnable

weight matrix at the $l$-th layer, $d'$ is the dimension of the latent representations, and $\sigma(\cdot)$ is the activation function. The final layer generates embeddings $\widetilde{\mathbf{Z}} \in \mathbb{R}^{\bar{N} \times d'}$. We then employ the $K$-means algorithm (Xu & Lange, 2019) to generate multiple prototypes $\bar{\mathbf{C}} \in \mathbb{R}^{\bar{O} \times d'}$, thereby obtaining representative samples from each cluster. Specifically, the Euclidean distance (Mafakheri et al., 2018) between the final embeddings of each node and the prototype of its corresponding subspace is calculated as:

$$f_d(\tilde{\mathbf{z}}_i, \bar{\mathbf{c}}_j) = \|\tilde{\mathbf{z}}_i - \bar{\mathbf{c}}_j\|_2, \qquad (2)$$

where $\tilde{\mathbf{z}}_i \in \widetilde{\mathbf{Z}}$ is the attribute vector of the $i$-th node, $\bar{\mathbf{c}}_j \in \bar{\mathbf{C}}$ is the $j$-th prototype. In this way, for each prototype, we select the top $n^k = k\bar{N}$ representative nodes approximated to the prototype, where $k$ is a hyperparameter that represents the ratio of samples selected. Subsequently, we collect the attribute matrix $\mathbf{B} \in \mathbb{R}^{n^k \bar{O} \times d}$ of these selected nodes.

Next, to project directed signals with clustering properties without directly sharing the raw data, we generate a masked matrix $\mathbf{R} \in \mathbb{R}^{n^k \bar{O} \times d}$ directly from Gaussian noise, which is hard to be reconstructed into raw graph data. After that, we utilize an MLP model (Liang et al., 2023; 2024; Liu et al., 2021a) as a projector to extract the meaningful clustering signal in a denoising-like manner, which makes $\mathbf{R}$ possibly approximate to $\mathbf{B}$:

$$\widehat{\mathbf{R}} = \mathrm{MLP}(\mathbf{R}; \bar{\theta}_{\mathrm{mlp}}), \qquad (3)$$

where $\widehat{\mathbf{R}} \in \mathbb{R}^{n^k \bar{O} \times d}$ is the reconstructed masked matrix, $\bar{\theta}_{\mathrm{mlp}}$ represents the parameters of the local MLP. The optimization objective is as follows:

$$\mathcal{L}_{\mathrm{mlp}} = \frac{1}{n^k \bar{O}}\|\mathbf{B} - \widehat{\mathbf{R}}\|_2. \qquad (4)$$

Finally, we upload $\mathbf{R}$ and the well-trained $\bar{\theta}_{\mathrm{mlp}}$ to the server in preparing for cross-subgraph link mending.

In summary, the proposed local model learning strategy offers two major advantages: it 1) collects and uploads key clustering signals that are hard to reconstruct into raw data, and 2) preserves more trustworthy clustering properties of each cluster within the client, serving as reliable signals for subsequent information sharing between cross-subgraphs.

### 3.3. Cross-Subgraph Link Mending

Since a complete graph is divided into multiple subgraphs, leading to missing links across subgraphs. Therefore, after the server receives the clustering signals from each client, we propose the cross-subgraph link mending strategy.

In the first step, to infer the global samples on the server, we adopt $\mathbf{R}$ and $\bar{\theta}_{\mathrm{mlp}}$ uploaded by each client, which ensures that these representative nodes are approximately to

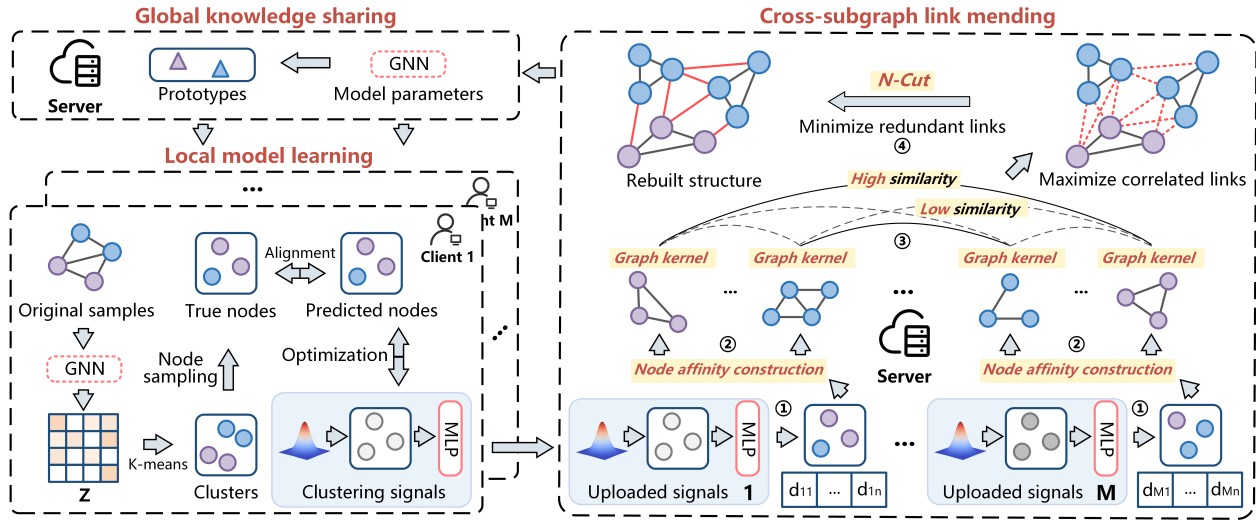

*Figure 2.* The framework of FedNCN is composed of three major components, i.e., the local model learning that collects and preserves more trustworthy clustering signals for destroyed sample connection restoration, the cross-subgraph link mending that establishes correct connections among subgraphs with the aid of prior learned clustering knowledge, and the global knowledge sharing that learns high-quality consensus features based on the mended graph and ensures reliable feedback to each client. Three parts are seamlessly integrated into a unified optimization framework. Notably, the cross-subgraph link mending is the core component that includes four steps, i.e., ① global sample inferring; ② intra-cluster link mending; ③ inter-cluster link mending; and ④ global structure refining.

**B.** Specifically, the server first optimizes the MLP projector based on obtaining multiple masked matrices and MLP model parameters. Then, we inference the global sample matrix $\widetilde{\mathbf{R}} \in \mathbb{R}^{n^k \bar{O} \times d}$ within the client by:

$$\widetilde{\mathbf{R}} = f_p(\mathbf{R}; \bar{\theta}_{\mathrm{mlp}}), \tag{5}$$

where $f_p(\cdot)$ denotes the projector function.

In the second step, after inferring the $\widetilde{\mathbf{R}}$, we cannot directly obtain the intra-cluster links due to privacy constraints. Therefore, we attempt to calculate the potential affinity between nodes within a cluster in a low-dimensional proximity manner (Yang et al., 2024). Specifically, we first divide the nodes belonging to the same cluster within each client into a single space. Then, we calculate the node similarity:

$$\mathrm{Sim}(\tilde{\mathbf{r}}_i, \tilde{\mathbf{r}}_j) = \frac{\tilde{\mathbf{r}}_i (\tilde{\mathbf{r}}_j)^\top}{||\tilde{\mathbf{r}}_i|| \cdot ||\tilde{\mathbf{r}}_j||}, \tag{6}$$

where $\mathrm{Sim}(\cdot, \cdot)$ is similarity function, $\tilde{\mathbf{r}}_i \in \widetilde{\mathbf{R}}$ and $\tilde{\mathbf{r}}_j \in \widetilde{\mathbf{R}}$ are attribute vector of $i$-th node and $j$-th node, respectively. Subsequently, we construct the refined subgraph $\mathcal{G}_{\mathrm{sub}}^o = \{\mathcal{V}_{\mathrm{sub}}^o, \mathcal{E}_{\mathrm{sub}}^o\}$ with $n^k$ nodes from each cluster by Eq. (6). Here, $\mathcal{E}_{\mathrm{sub}}^o$ is the edge set represented by the adjacency matrix $\mathbf{A}_{\mathrm{sub}}^o \in \mathbb{R}^{n^k \times n^k}$:

$$a_{ij}^o = \begin{cases} 1, & \text{if } (v_i^o, v_j^o) \in \mathcal{V}_{\mathrm{sub}}^o, \text{and}, v_j^o \in \mathcal{T} \\ 0, & \text{otherwise} \end{cases}, \tag{7}$$

where $\mathcal{T}$ is the target node set of the $i$-th node in the $\mathcal{G}_{\mathrm{sub}}^o$.

In the third step, to restore the cross-subgraph missing links as much as possible, we identify cross-subgraph relationships by the graph kernel method and then mend the cross-subgraph link in an approximately optimal manner (Borgwardt et al., 2020). Specifically, we first calculate the graph-level affinity matrix $\mathbf{S} \in \mathbb{R}^{\sum_{i=1}^M \bar{O}_i \times \sum_{i=1}^M \bar{O}_i}$ to measure relationships between subgraphs generated from clusters:

$$s_{ij} = \mathrm{GK}(\mathcal{G}_i^o, \mathcal{G}_j^o), \tag{8}$$

where $\mathrm{GK}(\cdot, \cdot)$ denotes graph kernel similarity function (e.g., COS (Choromanski, 2023), WL (Ju et al., 2023) and SP (Borgwardt et al., 2020)). See Appendix B for detailed descriptions of graph kernels. $s_{ij} \in \mathbf{S}$ represents the affinity between $\mathcal{G}_i^o$ and $\mathcal{G}_j^o$. Then, we construct edges between any two subgraph pairs to generate a global graph $\mathcal{G}^\phi = \{\mathcal{V}^o, \mathcal{E}^\phi\}$. Here, the total number of the $\mathcal{G}^\phi$ is $N^o = \sum_{i=1}^M k \bar{N}_i \bar{O}_i$, and the corresponding nodes set is $\mathcal{V}^o = \{v_1^o, v_2^o, \ldots, v_{N^o}^o\}$, the edges set is $\mathcal{E}^\phi$ represented by adjacency matrix $\mathbf{A}^\phi \in \mathbb{R}^{N^o \times N^o}$.

In the last step, we design an improved N-Cut method to ensure that the $\mathcal{G}^\phi$ focuses on the links that preserve the key structure, while eliminating redundant links. Specifically, we calculate the contribution of all cross-subgraph edges in the global graph as:

$$\mathrm{Con}(a_{ij}^\phi) = \frac{a_{ij}^\phi}{f_a(\mathcal{V}_1^o, \mathcal{V}^o)} + \frac{a_{ij}^\phi}{f_a(\mathcal{V}_2^o, \mathcal{V}^o)}, \tag{9}$$

where $\mathrm{Con}(\cdot)$ is the contribution function, $f_a(\mathcal{V}_1^o, \mathcal{V}^o) =$

**Algorithm 1** Training Procedure of FedNCN

---

    **Input:** Complete graph $\mathcal{G}$. For each subgraph: attribute matrix $\bar{\mathbf{X}}$; adjacency matrix $\bar{\mathbf{A}}$; server epoch $E_{\text{server}}$; client epoch $E_{\text{client}}$.
    **Output:** Clustering results $\mathbf{Y}$ at clients.
    Initialize $\bar{\mathbf{C}}^{(0)}$, $\theta_{\text{mlp}}^{(0)}$ and $\theta_{\text{gnn}}^{(0)}$ of all clients.
    **for** $i = 1$ **to** $E_{\text{server}}$ **do**
        `// Local model learning`
        **for** $j = 1$ **to** $E_{\text{client}}$ **do**
            Generate $\widetilde{\mathbf{Z}}$ for each client by Eq. (1).
        **end for**
        Obtain $\bar{\mathbf{C}}$ from $\widetilde{\mathbf{Z}}$ by $K$-means for each client.
        Obtain $\mathbf{B}$ with $\bar{\mathbf{C}}$ and $\widetilde{\mathbf{Z}}$ by Eq. (2).
        Generate $\mathbf{R}$ with a shape similar to $\mathbf{B}$ using noise.
        Train the MLP with $\mathbf{R}$ and $\mathbf{B}$ by Eqs. (3) and (4).
        Upload $\mathbf{R}$ and $\bar{\theta}_{\text{mlp}}$ to the server.
        `// Cross-subgraph link mending`
        Infer $\widetilde{\mathbf{R}}$ with $\mathbf{R}$ and $\bar{\theta}_{\text{mlp}}$ by Eq. (5).
        Generate the subgraph $\mathcal{G}_{\text{sub}}^{o}$ by Eqs. (6) and (7).
        Generate global graph $\mathcal{G}^{\phi}$ by Eq. (8).
        Refine the mended graph $\mathcal{G}^{\psi}$ by Eq. (9).
        `// Global knowledge sharing`
        Train global GNN model by Eqs. (10)-(12).
        Backpropagate the $\widetilde{\mathbf{C}}$ and $\tilde{\theta}_{\text{gnn}}$ to each client.
    **end for**
    Obtain clustering results $\mathbf{Y}$ at clients by $K$-means.
    **return** $\mathbf{Y}$

---

$\sum_{i \in \mathcal{V}_1^o, j \in \mathcal{V}^o} a_{ij}^{\phi}$ and $f_a(\mathcal{V}_2^o, \mathcal{V}^o) = \sum_{i \in \mathcal{V}_2^o, j \in \mathcal{V}^o} a_{ij}^{\phi}$ represent the association of the two subgraphs connected by edge $a_{ij}^{\phi}$ with the $\mathcal{G}^{\phi}$, respectively. $\mathcal{V}_1^o$ and $\mathcal{V}_2^o$ correspond to the node sets of two subgraphs within the $\mathcal{G}^{\phi}$. In this way, we retain the key edges based on their contribution, obtaining the mended graph $\mathcal{G}^{\psi}$ and its adjacency matrix $\mathbf{A}^{\psi} \in \mathbb{R}^{N^o \times N^o}$.

The merits of the proposed cross-subgraph link mending strategy can be summarized as: 1) it effectively leverages the prior learned clustering knowledge to enable the model to accurately conduct cross-subgraph link mending, and 2) a topological refinement approach that maximizes trustworthy connections while minimizing redundant ones improves the topology between subgraphs for better clustering.

### 3.4. Global Knowledge Sharing

After achieving the cross-subgraph links mending, we aim to learn the missing structure information from the $\mathcal{G}^{\psi} = \{\mathcal{V}^o, \mathcal{E}^{\psi}\}$ with $O$ cluster centers, where $\mathcal{E}^{\psi}$ is the edges set, $\mathbf{X}^{\psi} \in \mathbb{R}^{N^o \times d}$ denotes the attribute matrix, $\mathbf{A}^{\psi} \in \mathbb{R}^{N^o \times N^o}$ is the adjacency matrix. Inspired by previous work (Liu et al., 2025), the prototypes often retain the clustering properties. Consequently, we construct a GNN generator that is consistent with the local model at the server to learn

consensus prototypes $\widetilde{\mathbf{C}} \in \mathbb{R}^{O \times d'}$ and well-trained GNN parameters $\tilde{\theta}_{\text{gnn}}$, achieved by minimizing the reconstruction loss $\mathcal{L}_{\text{MSE}}$ and the clustering loss $\mathcal{L}_{\text{KL}}$ as:

$$\mathcal{L} = \mathcal{L}_{\text{MSE}} + \mathcal{L}_{\text{KL}}. \tag{10}$$

Here, $\mathcal{L}_{\text{MSE}}$ and $\mathcal{L}_{\text{KL}}$ are calculated by:

$$\mathcal{L}_{\text{MSE}} = \frac{1}{N^o}\|\widehat{\mathbf{X}}^{\psi} - \mathbf{X}^{\psi}\|_2, \tag{11}$$

$$\mathcal{L}_{\text{KL}} = \text{KL}(\mathbf{P}\|\mathbf{Q}) = \sum_i \sum_j p_{ij} \log \frac{p_{ij}}{q_{ij}}, \tag{12}$$

where $\widehat{\mathbf{X}}^{\psi} \in \mathbb{R}^{N^o \times d}$ is reconstructed node attributes matrix of the $\mathcal{G}^{\psi}$ by GNN decoder, the soft assignment distribution $\mathbf{Q} \in \mathbb{R}^{N^o \times O}$ of latent representations is calculated by the Student's $t$-distribution (Gong et al., 2022a), while the target distribution $\mathbf{P} \in \mathbb{R}^{N^o \times O}$ is obtained by sharpening $\mathbf{Q}$.

In summary, the proposed global knowledge-sharing strategy provides significant benefits: 1) with the aid of the mended topology information, the global model is enabled to learn consensus sample representations, which ensures reliable feedback to each client to learn clustering-friendly features; and 2) the distinct nature of the sub-model parameters, where the locally uploaded $\bar{\theta}_{\text{mlp}}$ and the globally returned $\tilde{\theta}_{\text{gnn}}$ are not identical, which further guarantees privacy security during data transmission.

### 3.5. Client-Server Collaborative Learning

After the global model learns the $\mathcal{G}^{\psi}$ on the server, each client receives the updated $\widetilde{\mathbf{C}}$ and $\tilde{\theta}_{\text{gnn}}$. Subsequently, $\widetilde{\mathbf{C}}$ is employed as the cluster center for the next round of local model training, and $\tilde{\theta}_{\text{gnn}}$ is utilized to initialize the parameters of the GNN. In this way, the discriminative ability of the local representations could be further enhanced. Finally, client-server collaborative learning is achieved by minimizing the total loss function of the local GNN (Similar to Eqs. (10)-(12)). Algorithm 1 provides a detailed description of the FedNCN learning process. The Detailed algorithms for the client and the server are provided in Appendix C.

## 4. Experiments

### 4.1. Experiment Setup

**Benchmark Datasets** Following the experimental setup from FedTAD (Zhu et al., 2024), we construct distributed subgraphs by dividing the dataset into 5 clients, 10 clients, and 20 clients, respectively, where each client has a subgraph that is part of a complete graph. Specifically, we use CiteSeer (Liu et al., 2023a), PubMed (Jiang et al., 2024), Amazon-Computer, Amazon-Photo (Lin et al., 2021), and Questions (Platonov et al., 2024) as our experimental benchmark datasets. For a detailed description of the datasets, please see Appendix D.1.

*Table 1.* Performance comparison across different federated node-level clustering methods. Notably, all compared methods are evaluated under unsupervised settings on five benchmark datasets to ensure a fair comparison.

| Methods | CiteSeer (5 Clients) | | | | CiteSeer (10 Clients) | | | | CiteSeer (20 Clients) | | | |
|---|---|---|---|---|---|---|---|---|---|---|---|---|
| | ACC | NMI | ARI | F1 | ACC | NMI | ARI | F1 | ACC | NMI | ARI | F1 |
| FedSage+* | 16.13±1.74 | 2.69±0.40 | 0.18±0.59 | 11.41±0.75 | 17.18±2.56 | 5.43±0.32 | -0.13±0.55 | 11.27±0.62 | 15.56±2.03 | 9.54±0.21 | 0.26±0.52 | 10.06±0.84 |
| FedPUB* | 16.80±3.40 | 0.00±0.00 | 0.00±0.00 | 4.11±0.56 | 9.05±5.17 | 0.00±0.00 | 0.00±0.00 | 3.05±1.62 | 11.17±6.48 | 0.00±0.00 | 0.00±0.00 | 3.75±2.04 |
| FedTAD* | 17.59±3.33 | 1.21±1.43 | 0.60±1.00 | 7.13±2.55 | 19.12±1.57 | 4.01±2.63 | 1.57±1.27 | 9.89±2.69 | 17.77±1.54 | 3.42±3.60 | 1.08±1.37 | 6.77±2.40 |
| FedGTA* | 24.05±0.80 | 5.68±1.06 | 3.54±0.70 | 19.95±1.05 | 26.34±0.61 | 8.36±1.60 | 4.74±1.95 | 19.67±0.71 | 28.05±1.62 | 12.35±2.08 | 4.32±2.08 | 17.13±2.21 |
| FedIIH* | 14.21±2.91 | 0.00±0.00 | 0.00±0.00 | 3.73±0.62 | 17.58±6.07 | 0.69±0.79 | 0.42±0.49 | 4.82±1.08 | 18.34±7.52 | 0.02±0.05 | -0.07±0.13 | 5.28±1.93 |
| **FedNCN** | **54.98±1.78** | **13.27±2.45** | **19.67±3.71** | **24.78±1.42** | **58.99±2.96** | **17.53±1.31** | **15.74±4.01** | **27.12±2.92** | **57.47±1.55** | **19.69±1.92** | **17.96±1.72** | **32.82±1.09** |

| Methods | PubMed (5 Clients) | | | | PubMed (10 Clients) | | | | PubMed (20 Clients) | | | |
|---|---|---|---|---|---|---|---|---|---|---|---|---|
| FedSage+* | 49.53±6.33 | 8.67±3.77 | 4.83±4.72 | 33.75±4.49 | 40.71±7.84 | 2.22±4.47 | 0.03±0.00 | 26.16±6.01 | 43.77±8.32 | 1.96±1.74 | -0.02±3.57 | 30.81±3.20 |
| FedPUB* | 31.97±9.44 | 0.00±0.00 | 0.00±0.00 | 14.07±3.16 | 32.03±9.04 | 0.00±0.00 | 0.00±0.00 | 14.00±3.19 | 28.30±9.25 | 0.00±0.00 | 0.00±0.00 | 12.43±3.30 |
| FedTAD* | 34.06±4.71 | 1.16±0.99 | 0.84±0.44 | 26.88±3.47 | 37.33±4.63 | 0.39±0.51 | 0.54±0.89 | 20.24±3.87 | 36.51±5.23 | 0.44±0.69 | 0.72±1.19 | 18.18±1.64 |
| FedGTA* | 46.22±2.50 | 6.15±1.93 | 6.33±1.90 | 39.60±2.77 | 46.87±1.66 | 2.08±0.83 | 1.79±1.19 | 33.27±1.92 | 48.75±1.70 | 2.52±0.73 | 2.63±0.83 | 32.24±0.97 |
| FedIIH* | 32.30±9.71 | 0.04±0.09 | 0.13±0.27 | 14.55±3.53 | 33.55±7.77 | 0.02±0.03 | 0.02±0.03 | 14.64±2.84 | 28.30±9.25 | 0.00±0.00 | 0.00±0.00 | 12.43±3.30 |
| **FedNCN** | **63.42±1.59** | **10.53±3.48** | **13.85±3.57** | **44.03±2.30** | **64.95±1.48** | **10.58±1.75** | **13.59±4.56** | **44.37±3.45** | **66.33±1.72** | **7.46±0.87** | **11.02±1.21** | **41.28±1.31** |

| Methods | Amazon-Computer (5 Clients) | | | | Amazon-Computer (10 Clients) | | | | Amazon-Computer (20 Clients) | | | |
|---|---|---|---|---|---|---|---|---|---|---|---|---|
| FedSage+* | 21.42±0.94 | 5.08±2.78 | 3.98±3.12 | 14.85±2.06 | 23.04±0.84 | 8.19±0.88 | 6.33±1.04 | 17.39±0.18 | 20.65±1.28 | 7.08±1.36 | 4.73±2.00 | 21.16±0.93 |
| FedPUB* | 11.74±12.96 | 0.00±0.00 | 0.00±0.00 | 2.29±2.40 | 22.39±12.26 | 0.00±0.00 | 0.00±0.00 | 4.95±3.17 | 14.25±12.14 | 0.00±0.00 | 0.00±0.00 | 4.02±3.46 |
| FedTAD* | 9.82±4.38 | 3.19±1.73 | 3.34±1.64 | 3.26±1.47 | 12.84±12.41 | 3.68±2.03 | 4.71±2.59 | 3.39±1.93 | 17.79±11.26 | 5.08±1.34 | 3.99±0.38 | 3.82±1.44 |
| FedGTA* | 21.40±1.76 | 12.07±3.95 | 8.38±3.27 | 8.96±1.01 | 20.96±1.46 | 6.46±1.22 | 3.71±1.03 | 6.92±0.52 | 22.08±1.25 | 7.25±0.96 | 3.46±0.76 | 6.30±0.28 |
| FedIIH* | 13.52±7.09 | 0.00±0.00 | 0.00±0.00 | 2.62±1.54 | 7.81±4.23 | 0.09±0.11 | 0.06±0.09 | 1.87±0.94 | 7.58±4.58 | 0.02±0.04 | 0.05±0.10 | 2.38±1.82 |
| **FedNCN** | **64.17±1.19** | **23.47±1.01** | **21.62±2.28** | **24.52±2.13** | **71.21±2.04** | **21.33±3.11** | **23.76±4.54** | **24.00±0.81** | **70.20±2.61** | **21.64±1.45** | **22.86±3.20** | **32.54±0.77** |

| Methods | Amazon-Photo (5 Clients) | | | | Amazon-Photo (10 Clients) | | | | Amazon-Photo (20 Clients) | | | |
|---|---|---|---|---|---|---|---|---|---|---|---|---|
| FedSage+* | 32.45±1.90 | 10.95±2.26 | 11.43±2.73 | 18.73±1.11 | 34.00±1.25 | 11.10±1.23 | 9.00±1.76 | 18.47±0.62 | 36.75±0.97 | 11.28±0.24 | 9.80±0.34 | 26.52±0.54 |
| FedPUB* | 14.89±7.63 | 0.00±0.00 | 0.00±0.00 | 3.44±1.74 | 8.42±3.17 | 0.00±0.00 | 0.00±0.00 | 1.91±0.49 | 12.71±6.81 | 15.20±0.00 | 15.20±0.00 | 6.02±3.58 |
| FedTAD* | 13.50±4.99 | 4.11±6.80 | 4.03±6.80 | 3.67±2.53 | 13.07±3.59 | 5.42±2.22 | 5.48±2.50 | 4.02±1.66 | 11.78±4.96 | 8.34±1.04 | 8.63±1.21 | 4.25±1.35 |
| FedGTA* | 35.54±1.96 | 14.30±4.04 | 11.88±4.09 | 11.99±1.15 | 32.68±1.12 | 11.84±0.76 | 9.59±1.13 | 10.83±0.59 | 29.40±1.17 | 10.82±0.75 | 7.84±0.87 | 9.60±0.34 |
| FedIIH* | 14.33±4.20 | 0.00±0.00 | 0.00±0.00 | 3.09±1.03 | 11.66±7.77 | 0.00±0.00 | 0.00±0.00 | 2.74±1.93 | 12.11±7.78 | 15.25±0.10 | 15.22±0.03 | 4.14±3.48 |
| **FedNCN** | **74.05±2.95** | **37.22±5.34** | **43.05±7.02** | **29.11±2.81** | **73.48±1.78** | **29.16±2.13** | **31.11±3.43** | **27.60±1.07** | **66.09±2.06** | **19.85±1.62** | **16.72±2.01** | **32.46±0.92** |

| Methods | Questions (5 Clients) | | | | Questions (10 Clients) | | | | Questions (20 Clients) | | | |
|---|---|---|---|---|---|---|---|---|---|---|---|---|
| FedSage+* | 78.08±2.10 | 0.96±1.51 | 1.31±1.51 | 39.24±0.80 | 76.31±1.50 | 0.84±1.70 | 0.82±2.18 | 38.29±1.24 | 76.00±0.63 | 0.64±0.72 | 0.57±0.96 | 36.25±0.76 |
| FedPUB* | 59.40±46.04 | 0.00±0.00 | 0.00±0.00 | 30.70±22.69 | 59.41±46.12 | 0.00±0.00 | 0.00±0.00 | 30.68±22.74 | 40.59±45.97 | 0.01±0.02 | -0.04±0.07 | 21.45±22.66 |
| FedTAD* | 56.05±27.81 | 0.18±0.13 | -0.83±0.78 | 35.82±12.35 | 66.30±31.30 | 0.10±0.06 | -0.28±0.32 | 38.57±14.78 | 69.06±34.36 | 0.06±0.07 | -0.32±0.41 | 38.74±12.99 |
| FedGTA* | 88.47±4.45 | 0.62±0.33 | 3.17±1.64 | 50.50±1.77 | 88.07±3.04 | 0.38±0.06 | 1.29±0.44 | 49.56±0.52 | 86.69±2.39 | 0.72±0.69 | 0.67±2.12 | 47.81±1.47 |
| FedIIH* | 77.40±6.22 | 0.00±0.00 | 0.00±0.00 | 30.70±22.69 | 78.26±3.44 | 0.00±0.00 | 0.00±0.00 | 30.68±22.74 | 80.06±2.77 | 0.38±0.73 | 0.63±1.33 | 41.63±2.85 |
| **FedNCN** | **94.64±2.04** | **1.88±0.86** | **6.44±2.93** | **52.89±1.73** | **91.27±4.69** | **1.64±0.82** | **4.62±2.34** | **51.41±2.15** | **93.03±5.14** | **1.08±0.81** | **2.93±3.28** | **50.80±2.29** |

\* Note that these supervised node-level FGL methods are adapted to the unsupervised scenario.

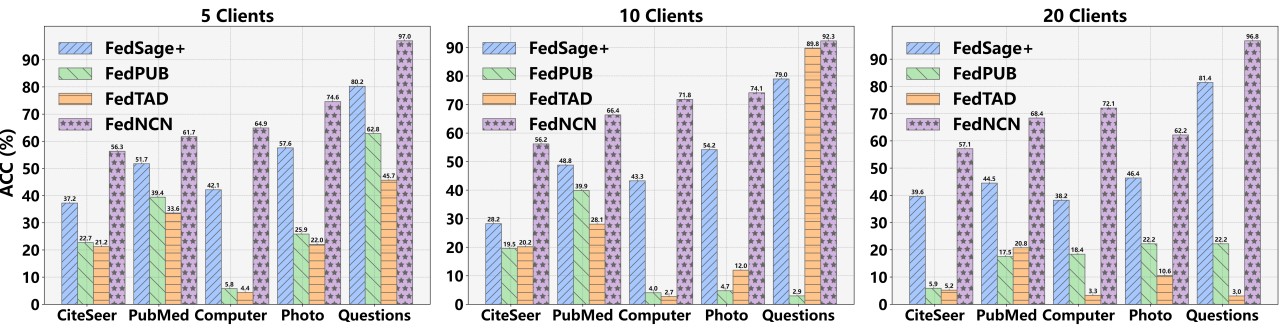

*Figure 3.* Performance comparison of FedNCN with three advanced federated node-level classification methods on five graph datasets.

**Baseline Methods** To demonstrate the superiority of Fed-NCN, we conduct comprehensive comparisons with two groups of baseline methods. The first group consists of five advanced FGL methods, namely FedSage+ (Zhang et al., 2021), FedPUB (Baek et al., 2023), FedTAD (Zhu et al., 2024), FedGTA (Li et al., 2024), and FedIIH (Yu et al., 2025). The second group comprises three classical FL aggregation strategies, including: FedAvg (McMahan et al., 2017), FedProx (Li et al., 2020), and FedPer (Arivazhagan et al., 2019), where the clients utilize our proposed local model to ensure fair and consistent comparison. We provide detailed descriptions of baseline methods in Appendix D.1.

*Table 2.* The performance of different FL aggregation strategies in federated node-level clustering.

| Datasets | Methods | ACC | NMI | ARI | F1 | avg. $\Delta$ |
|---|---|---|---|---|---|---|
| CiteSeer (10 Clients) | Local | 44.1±1.7 | 8.6±1.2 | 4.1±2.1 | 23.5±0.5 | - |
| | FedAvg | 55.9±2.8 | 4.2±0.5 | 0.4±0.8 | 16.5±0.7 | -0.8 |
| | FedProx | 45.0±3.0 | 8.1±0.9 | 3.3±1.3 | 23.7±0.7 | -0.1 |
| | FedPer | 48.7±1.0 | 7.8±2.3 | 4.7±3.0 | 22.8±1.0 | +0.9 |
| | **FedNCN** | **59.0±3.0** | **17.5±1.3** | **15.7±4.0** | **27.1±2.9** | **+9.7** |
| PubMed (10 Clients) | Local | 54.4±1.5 | 8.0±1.9 | 7.1±1.9 | 37.0±1.0 | - |
| | FedAvg | 63.7±2.9 | 5.2±1.7 | 4.8±0.7 | 23.9±6.9 | -2.2 |
| | FedProx | 56.5±0.9 | 2.1±0.6 | 3.2±0.6 | 35.3±0.7 | -2.1 |
| | FedPer | 55.5±1.9 | 7.5±1.0 | 6.7±1.6 | 36.8±1.2 | 0.0 |
| | **FedNCN** | **65.0±1.5** | **10.6±1.8** | **13.6±4.6** | **44.4±3.5** | **+6.8** |
| Amazon-Computer (10 Clients) | Local | 45.9±0.3 | 4.8±0.1 | 1.9±1.0 | 15.7±0.3 | - |
| | FedAvg | 67.1±0.0 | 0.8±0.0 | -0.2±0.0 | 13.5±0.0 | +3.2 |
| | FedProx | 43.1±3.0 | 2.8±0.3 | 0.7±0.7 | 15.9±0.4 | -1.5 |
| | FedPer | 50.2±1.1 | 2.5±0.7 | 1.1±1.0 | 15.7±0.5 | +0.3 |
| | **FedNCN** | **71.2±2.0** | **21.3±3.1** | **23.8±4.5** | **24.0±0.8** | **+18.0** |
| Amazon-Photo (10 Clients) | Local | 48.2±0.8 | 5.7±0.5 | 2.4±0.9 | 15.8±0.5 | - |
| | FedAvg | 63.5±1.3 | 3.1±0.8 | 1.6±0.3 | 15.0±0.6 | +2.8 |
| | FedProx | 49.7±1.8 | 5.3±0.1 | 4.2±1.2 | 16.9±0.3 | +1.0 |
| | FedPer | 57.0±1.4 | 5.3±0.8 | 4.7±1.0 | 17.7±0.7 | +3.2 |
| | **FedNCN** | **73.5±1.8** | **29.2±2.1** | **31.1±3.4** | **27.6±1.1** | **+22.4** |
| Questions (10 Clients) | Local | 78.1±1.8 | 0.5±0.1 | -0.7±0.7 | 45.8±0.2 | - |
| | FedAvg | 87.6±0.8 | 0.0±0.0 | 0.0±0.0 | 46.2±0.0 | +2.5 |
| | FedProx | 84.4±1.3 | 0.3±0. | 1.9±0.2 | 49.2±0.3 | +3.0 |
| | FedPer | 82.2±2.1 | 0.3±0.1 | 1.6±0.7 | 48.4±0.9 | +2.1 |
| | **FedNCN** | **91.3±4.7** | **1.6±0.8** | **4.6±2.3** | **51.4±2.2** | **+6.2** |

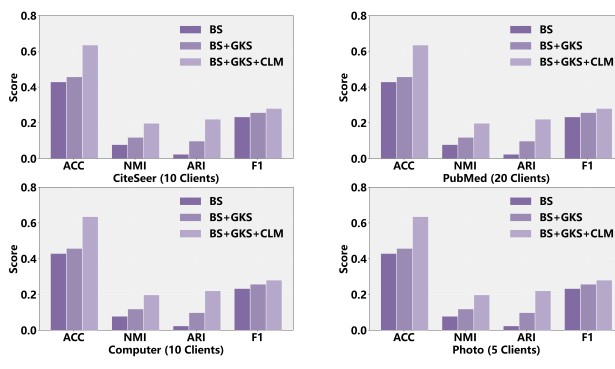

*Figure 4.* Ablation studies for CLM strategy and GKS strategy in federated node-level clustering scenarios.

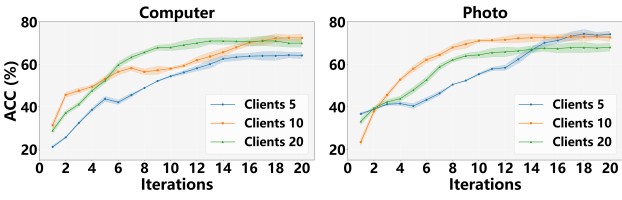

*Figure 5.* Algorithm convergence on Computer and Photo datasets.

**Implementation Details** To ensure experimental fairness, all methods are evaluated under identical hardware and settings. We utilize a four-layer GNN on both the client and the server to obtain node embeddings, with hidden layer dimensions are 500-500-2000-10. Moreover, we use a one-layer MLP to obtain the local clustering signals, which are then uploaded to the server. During model optimization, we adopt the Adam optimizer (Xiao et al., 2024) with a learning rate of 1e-3. The client-server interaction is conducted 20 times, with the local model training 10 epochs during each interaction. All methods are implemented using PyTorch 2.4.0 and a single NVIDIA GeForce RTX 4090 GPU.

**Clustering Metrics** To comprehensively evaluate the clustering performance of all methods, we adopt four widely used evaluation metrics, including Accuracy (ACC) (Liu et al., 2022b; Wang et al., 2022b;a;c; Cai et al., 2022), Normalized Mutual Information (NMI) (Liu et al., 2021b; 2022a; Wang et al., 2021c; Liang et al., 2023), Adjusted Rand Index (ARI) (Yang et al., 2023; Tu et al., 2022), and F1 Score (F1) (Liu et al., 2023b; Tu et al., 2024c;b; 2025a;b) to test clustering results from multiple perspectives.

### 4.2. Experimental Results

**Comparison in Unsupervised Learning Settings** To validate the effectiveness of the proposed FedNCN, which is the first federated node-level clustering framework with cross-subgraph link mending. We extend five advanced federated node-level classification methods into the corresponding graph clustering versions to ensure a fair comparison. To reduce randomness, each experiment is repeated five times, and the mean and standard deviation of the four clustering metrics are presented in Table 1. We can observe that, under different client partitions, FedNCN consistently outperforms the compared methods on five datasets. The reasons behind this are two-fold: 1) these approaches such as FedSage+, FedTAD, FedPUB, FedGTA, and FedIIH are specifically designed to enhance the quality of local-global interactions for better classification. As a result, their effectiveness is limited when dealing with unlabeled graph data; and 2) on the one hand, FedNCN is tailored for node-level clustering scenarios. On the other hand, we design a cross-subgraph link mending strategy that restores missing links on the server while not directly sharing the raw data, enabling the global model to generate clustering-friendly prototypes for better clustering.

**Comparison with Supervised FGL Methods** To further verify the effectiveness of FedNCN, we compare it with five representative supervised FGL methods: FedSage+, FedTAD, FedPUB, FedGTA, and FedIIH. Each method of these methods is trained with 5% of the labeled data. As shown in Fig. 3, the experimental results indicate that, under different client partitions, FedNCN consistently maintains strong competitiveness in clustering tasks, even when compared to federated node-level classification tasks (i.e., FedSage+,

FedTAD, and FedPUB) with limited labeled data, further highlighting the effectiveness of FedNCN. The comparative results of other methods can be found in Appendix D.2.

**Comparison of FL Aggregation Strategies** To verify the effectiveness of the global aggregation strategy, we compare FedNCN with the classical FL aggregation strategies. In our setup, local models are utilized to generate clustering signals, which are subsequently uploaded for global sharing via FL aggregation strategies including FedAvg, FedProx, and FedPer. We report the average of four clustering metrics and their average gains compared to local model testing in Table 2. As seen, compared to these FL aggregation strategies, FedNCN achieves noticeable improvements in clustering performance. For example, it achieves average improvements of 9.7%, 6.8%, 18.0%, 22.4%, and 6.2% on the CiteSeer, PubMed, Amazon-Computer, Amazon-Photo, and Questions datasets, respectively. These findings indicate that the integration of our global aggregation strategy significantly enhances its generalization ability across different datasets. In addition, for the federated node-level clustering tasks with 5 clients and 20 clients, the detailed experimental results can be found in Appendix D.2.

**Analysis of CLM Strategy and GKS Strategy** In this part, we evaluate the effectiveness of the proposed Cross-subgraph Link Mending (CLM) strategy and GKS (Global Knowledge Sharing) strategy. In our setup, "BS" denotes the local model of FedNCN. "BS+GKS" denotes the Fed-NCN variants with the CLM strategy being removed, and "BS+GKS+CLM" denotes the FedNCN. As summarized in Fig. 4, we can observe that 1) the four clustering metrics of "BS+GKS+CLM" are consistently better than both the "BS+GKS" and "BS"; 2) compared with the "BS", the performance of "BS+GKS" has a slight improvement. These findings can be attributed to the following points. Firstly, the cross-subgraph link mending strategy effectively restores the destroyed links on the server, thus facilitating better information sharing across clients; Secondly, the updated prototypes and model parameters obtained through the global knowledge-sharing strategy guide each local model to generate a robust target distribution.

**Convergence Analysis** In Fig. 5, we plot the ACC metric during the training iterations to monitor the model convergence on five random experiments on the Amazon-Photo and Amazon-Computer datasets to evaluate the robustness of the model. Under different client settings, the ACC metric consistently converged to a stable value within the 20 iterations, with only minor fluctuations. This further highlights the advantages of FedNCN in handling node-level clustering tasks.

**Analysis of Hyper-Parameter $k$** FedNCN introduces a hyper-parameter $k$, leveraged to determine the ratio of key clustering samples uploaded to the server from each cluster.

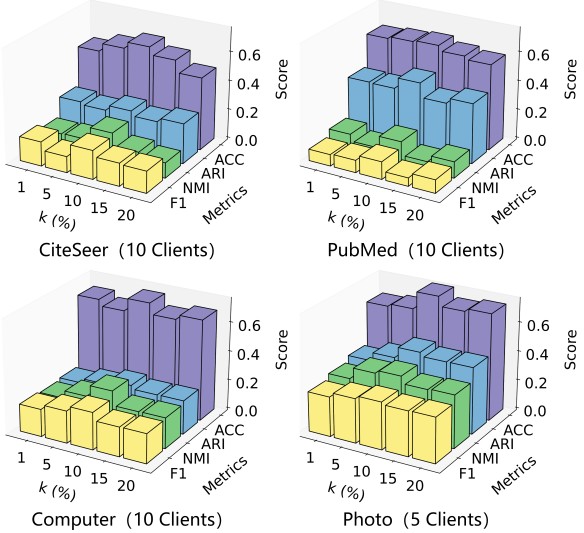

*Figure 6.* The sensitivity of FedNCN with the variation of a hyper-parameter $k$ on CiteSeer, PubMed, Computer, and Photo datasets.

We conduct experiments on four datasets to analyze the impact of this $k$ by varying its value from 1% to 20%. From the results in Fig. 6, we can observe that 1) the hyperparameter $k$ indeed plays a crucial role in FedNCN, suggesting that searching $k$ from a reasonable hyper-parameter region could benefit clustering performance; 2) uploading a few samples may result in sub-optimal performance, as it fails to reconstruct a meaningful global graph structure, which reduces the quality of the model collaborative learning; and 3) uploading too many samples may hinder clustering performance due to excessive noise, leading to unreliable links mending on the server.

## 5. Conclusion

In this paper, we explore a new research task, federated node-level clustering with cross-subgraph link mending, which is the first attempt to restore the destroyed links between clients due to graph partition under unsupervised settings. To tackle this task, we propose a novel unsupervised node-level FGL framework, called FedNCN. In our method, the proposed local model learning method, the cross-subgraph link mending strategy, and the global knowledge-sharing strategy collectively succeed in restoring missing links without reliance on annotations, thereby facilitating the information negotiation among multiple clients for better clustering. Extensive experimental results across five graph datasets demonstrate that FedNCN can effectively mend missing links caused by sample partition in an unsupervised setting. In future work, we plan to enhance the generalization capabilities of FedNCN, extending its application to a broader

range of FGL tasks, such as incomplete federated graph clustering and federated multi-view clustering.

## Acknowledgement

This work is supported by the Key Research and Development Program of Hainan Province (Grant No. ZDYF2024GXJS014, ZDYF2023GXJS163), the National Natural Science Foundation of China (Grant No. 62162022, 62162024), the Natural Science Foundation of Hainan University (Grant No. XJ2400009401), the Youth Foundation Project of Hainan Natural Science Foundation (Grant No. 624QN279), and the Innovative research project for Graduate students in Hainan Province (Grant No. Qhyb2023-28).

## Impact Statement

This paper presents work whose goal is to advance the field of Machine Learning. There are many potential societal consequences of our work, none which we feel must be specifically highlighted here.

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

## A. Notations

In this section, we provide all notations of the proposed FedNCN are summarized in Table 3.

*Table 3.* Basic notations for the proposed FedNCN.

| Notations | Meaning | Notations | Meaning |
|---|---|---|---|
| $\bar{\mathbf{X}} \in \mathbb{R}^{\bar{N} \times d}$ | Attribute matrix of each subgraph | $\bar{\mathbf{A}} \in \mathbb{R}^{\bar{N} \times \bar{N}}$ | Original adjacency matrix of each subgraph |
| $\bar{\mathbf{D}} \in \mathbb{R}^{\bar{N} \times \bar{N}}$ | Degree matrix of each subgraph | $\widetilde{\mathbf{A}} \in \mathbb{R}^{\bar{N} \times \bar{N}}$ | Normalized adjacency matrix of each subgraph |
| $\mathbf{I} \in \mathbb{R}^{\bar{N} \times \bar{N}}$ | Identity matrix of each subgraph | $\mathbf{Z} \in \mathbb{R}^{\bar{N} \times d'}$ | Latent representations of each subgraph |
| $\widetilde{\mathbf{Z}} \in \mathbb{R}^{\bar{N} \times d'}$ | Final latent representations of each subgraph | $\bar{\mathbf{C}} \in \mathbb{R}^{\bar{O} \times d'}$ | Prototypes of each subgraph |
| $\mathbf{B} \in \mathbb{R}^{n^k \bar{O} \times d}$ | Truth samples attribute matrix of each subgraph | $\mathbf{R} \in \mathbb{R}^{n^k \bar{O} \times d}$ | Masked matrix of each subgraph |
| $\widehat{\mathbf{R}} \in \mathbb{R}^{n^k \bar{O} \times d}$ | Reconstructed masked matrix of each subgraph | $\widetilde{\mathbf{R}} \in \mathbb{R}^{n^k \bar{O} \times d}$ | Global samples attribute matrix of each subgraph |
| $\mathbf{S} \in \mathbb{R}^{\sum_{i=1}^{M} \bar{O}_i \times \sum_{i=1}^{M} \bar{O}_i}$ | Graph-level affinity matrix | $\mathbf{Q} \in \mathbb{R}^{N^o \times O}$ | Soft assignment distribution of the mended graph |
| $\mathbf{X}^\psi \in \mathbb{R}^{N^o \times d}$ | Attribute matrix of the mended graph | $\mathbf{P} \in \mathbb{R}^{N^o \times O}$ | Target distribution of the mended graph |
| $\mathbf{A}^\psi \in \mathbb{R}^{N^o \times N^o}$ | Adjacency matrix of the mended graph | $\widetilde{\mathbf{C}} \in \mathbb{R}^{O \times d'}$ | Consensus prototypes |

## B. Different Graph Kernel Methods

In this section, we provide a detailed description of different graph kernel methods. Common graph kernel methods include the cosine kernel, the weisfeiler-lehman kernel, the shortest path kernel, and the lovász-theta kernel. The specific descriptions are as follows:

- **The Cosine (COS) Kernel** (Choromanski, 2023) calculates the similarity by compressing each graph as a feature vector and then using cosine to measure the angle between these feature vectors. This method combines the graph structure and the node attributes.

- **The Weisfeiler-Lehman (WL) Kernel** (Ju et al., 2023) calculates the similarity between the two graphs by re-marking the nodes of the graph through iteration, thus generating the hierarchical structure representations.

- **The Shortest Path (SP) Kernel** (Borgwardt et al., 2020) calculates the similarity between the two graphs by evaluating the distribution of shortest path lengths between corresponding nodes, which captures the global structure information of the graphs.

- **The Lovász-Theta (LT) Kernel** (Johansson et al., 2014) calculates the similarity between the two graphs by solving Lovász theta function, which reflects the size of the largest independent set in the graph.

## C. Algorithms

In this section, we provide the algorithms for the local model learning method executed on the client, the cross-subgraph link mending strategy, and the global knowledge-sharing strategy on the server in our proposed FedNCN. Specifically, the process executed on the client is shown in Algorithm 2, while the process executed on the server is presented in Algorithm 3.

## D. Experiments

In this section, we first provide a detailed description of the five benchmark datasets used in Subsection D.1. Then, we elaborately introduce the baseline methods and our proposed FedNCN method. Finally, we further report detailed information on the comparative experiments and ablation studies of the FedNCN method, as in Subsection D.2.

### D.1. Experiment Setup

**Datasets** We report the detailed information of five benchmark datasets, as shown in Table 4.

CiteSeer and PubMed (Jiang et al., 2024) are both used for citation graphs. CiteSeer is a dataset containing academic papers and their citation relationships, primarily used for analyzing citation networks and is especially suitable for node-level graph

---

**Algorithm 2** FedNCN for Client Algorithm

---

**Input:** Complete graph $\mathcal{G}$; attribute matrix $\bar{\mathbf{X}}$; adjacency matrix $\bar{\mathbf{A}}$; GNN epoch $E_{\mathrm{gnn}}$; MLP epoch $E_{\mathrm{mlp}}$.

Initialize model parameters $\bar{\mathbf{C}}^{(0)}$, $\theta_{\mathrm{mlp}}^{(0)}$ and $\theta_{\mathrm{gnn}}^{(0)}$ from server.

**Output:** Masked matrix $\mathbf{R}$ and MLP parameters $\bar{\theta}_{\mathrm{mlp}}$.

**for** $j = 1$ to $E_{\mathrm{gnn}}$ **do**

    Generate $\widetilde{\mathbf{Z}}$ by Eq. (1).

**end for**

Obtain $\bar{\mathbf{C}}$ from $\widetilde{\mathbf{Z}}$ by $K$-means.

Obtain $\mathbf{B}$ with $\bar{\mathbf{C}}$ and $\widetilde{\mathbf{Z}}$ by Eq. (2).

Generate $\mathbf{R}$ with a shape similar to $\mathbf{B}$ using gaussian noise.

**for** $j = 1$ to $E_{\mathrm{mlp}}$ **do**

    Obtain $\widehat{\mathbf{R}}$ from $\mathbf{R}$ by Eq. (3).

    Optimize the local MLP model by minimizing Eq. (4).

**end for**

Upload $\mathbf{R}$ and $\bar{\theta}_{\mathrm{mlp}}$ to the server.

---

**Algorithm 3** FedNCN for Server Algorithm

---

**Input:** Masked matrix $\mathbf{R}$; MLP parameters $\bar{\theta}_{\mathrm{mlp}}$; GNN epoch $E_{\mathrm{gnn}}$.

**Output:** Consensus prototype $\widetilde{\mathbf{C}}$ and well-trained GNN parameters $\tilde{\theta}_{\mathrm{gnn}}$.

Infer $\widetilde{\mathbf{R}}$ with $\mathbf{R}$ and $\bar{\theta}_{\mathrm{mlp}}$ by Eq. (5).

Generate the refined subgraph $\mathcal{G}_{\mathrm{sub}}^{o}$ by Eqs. (6) and (7).

Calculate the graph-level similarity $\mathbf{S}$ by Eq. (8).

Generate the global graph $\mathcal{G}^{\phi}$ with $\mathbf{S}$.

Generate the mended graph $\mathcal{G}^{\psi}$ by Eq. (9).

**for** $i = 1$ to $E_{\mathrm{gnn}}$ **do**

    Calculate the global reconstruction loss $\mathcal{L}_{\mathrm{MSE}}$ by Eq. (11).

    Calculate the global clustering loss $\mathcal{L}_{\mathrm{KL}}$ by Eq. (12).

    Optimize the GNN model by minimizing Eq. (10).

**end for**

Obtain consensus prototype $\widetilde{\mathbf{C}}$ and well-trained GNN parameters $\tilde{\theta}_{\mathrm{gnn}}$.

Share $\widetilde{\mathbf{C}}$ and $\tilde{\theta}_{\mathrm{gnn}}$ to each client.

---

learning tasks. Each node represents each paper, and the edges represent the citation relationships between papers. Similarly, the PubMed dataset is an academic citation dataset sourced from the PubMed database, containing literature from the fields of medicine. PubMed dataset is mainly suitable for training and evaluating large-scale node-level graph learning models.

Amazon-Computer and Amazon-Photo (Lin et al., 2021) datasets are part of the Amazon product graph. Amazon-Computer dataset focuses on products related to computers and is part of the Amazon product review dataset. Each node represents a product (related to computers), and the edges represent the correlation between products. Amazon-Photo dataset is another subset of the Amazon product review dataset, focusing on products related to photos and cameras. Similar to the Computer dataset, each node represents a product (related to photos and cameras), and the edges represent the correlation between products. Both of these datasets are commonly used for node-level graph learning tasks.

Questions dataset (Platonov et al., 2024) is commonly used in the field of graph learning, especially in applications such as question recommendation systems or question graph analysis. In these applications, questions are abstracted as nodes in the graph, and the correlation between questions is represented by the edges in the graph.

**Baselines and Our Model**

- **FedSage+:** This method (Zhang et al., 2021) designs a missing neighbor generator based on FedSage to compensate for the missing graph structure information. In this process, each client first receives the node representations from the other clients and then computes the gradient of the distance between these representations and local node features. Finally, the gradient is used to update the graph generator.

*Table 4.* Dataset summary.

| Dataset | Nodes | Edges | Dimensions | Clusters | Type | Domian |
|---------|-------|-------|-----------|----------|------|--------|
| CiteSeer | 3327 | 9104 | 3703 | 6 | Graph | Citation |
| PubMed | 19717 | 88648 | 500 | 3 | Graph | Citation |
| Amazon-Computer | 13752 | 491722 | 767 | 10 | Graph | Computer Product |
| Amazon-Photo | 7650 | 238162 | 754 | 8 | Graph | Photo Product |
| Questions | 48921 | 153540 | 301 | 2 | Graph | Q&A System |

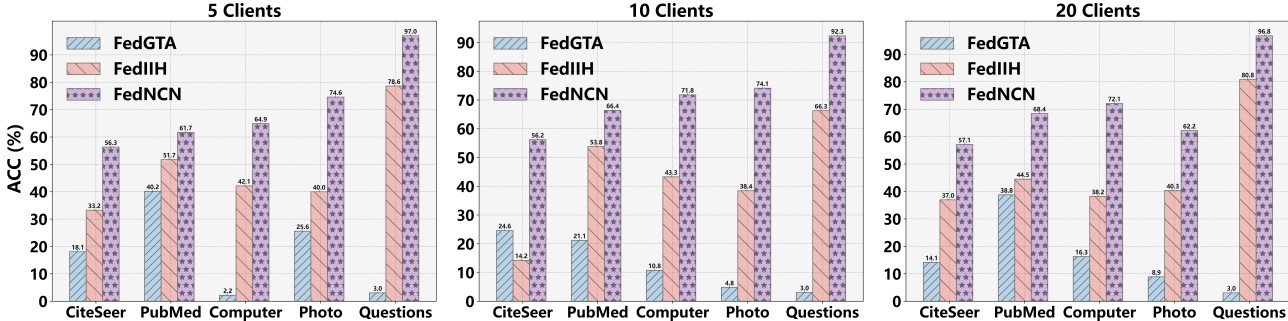

*Figure 7.* Performance comparison of FedNCN with two advanced federated node-level classification methods on five graph datasets.

- **FedPUB:** This method (Baek et al., 2023) proposes a novel personalized subgraph FL framework. In this approach, the central server constructs a random graph under the guidance of real labels and distributes it to each client. Subsequently, each client learns this random graph to obtain the functional embeddings, which are then uploaded to the server for identifying the community consisting of a group of densely connected subgraphs. Moreover, FedPUB learns a personalized sparse mask on each client to select only the relevant subset of the subgraph for parameter updates, enhancing the aggregation performance.

- **FedTAD:** This method (Zhu et al., 2024) is a topology-aware subgraph federated learning framework that addresses subgraph heterogeneity by enhancing the knowledge transfer from local models to the global model. Each client first decouples node and structure variation, revealing their difference in structure homophily and label distribution. Then, a knowledge distillation process is applied to transfer reliable local knowledge to the global model, which optimizes the performance of the local model.

- **FedGTA:** This method (Li et al., 2024) is a novel optimization strategy for FGL, which is designed to address scalability and performance issues in existing FGL methods when dealing with large-scale graph data. This method introduces a topology-aware mechanism that uses a mixed aggregation of neighbor features to enhance the capacity of the local model to capture graph structure information.

- **FedIIH:** This method (Yu et al., 2025) is a federated learning framework that addresses both inter- and intra-subgraph heterogeneity. It uses a hierarchical variational model to capture inter-subgraph similarities and then disentangles subgraphs into multiple latent factors to handle intra-heterogeneity, which improves the overall model robustness.

- **FedNCN:** This is our Federated Node-level Clustering Network with inter-client subgraph mending. On the client, an MLP-based projector is designed to implicitly obtain clustering signals, which are then uploaded for subsequent inter-client link mending. On the server, the cross-subgraph link mending strategy is proposed to maximize correlated links through graph similarity estimation and minimize redundant links by using a method similar to N-Cut. Subsequently, a GNN-based generator is developed to learn the consensus prototypes from this mended graph, which enhances the clustering performance of each client.

- **Local:** This method is a non-FL approach, which is the proposed FedNCN local model.

*Table 5.* The performance of different federated learning strategies in federated node-level clustering. Here, "Local" denotes the use of only our local model, while in "FedAvg", "FedProx", and "FedPer", different aggregation strategies are applied by the server, with the client all using our local model.

| Datasets | Methods | ACC | NMI | ARI | F1 | avg. $\Delta$ | Datasets | Methods | ACC | NMI | ARI | F1 | avg. $\Delta$ |
|---|---|---|---|---|---|---|---|---|---|---|---|---|---|
| CiteSeer (5 Clients) | Local | 43.5±2.8 | 6.5±2.2 | 5.1±4.2 | 21.1±1.8 | - | CiteSeer (20 Clients) | Local | 45.5±0.6 | 10.8±0.8 | 4.0±1.0 | 28.0±1.0 | - |
| | FedAvg | 53.3±2.2 | 3.9±1.2 | 1.9±1.7 | 16.6±1.6 | -0.1 | | FedAvg | 53.9±1.1 | 8.0±0.8 | 1.4±0.8 | 22.3±1.5 | -0.7 |
| | FedProx | 41.2±1.9 | 4.9±1.0 | 3.2±1.5 | 21.0±0.7 | -1.5 | | FedProx | 48.3±3.6 | 10.1±0.6 | 2.8±0.3 | 27.1±1.1 | 0 |
| | FedPer | 53.1±1.8 | 7.3±2.2 | 8.6±2.5 | 21.6±1.2 | +3.6 | | FedPer | 47.8±1.3 | 11.0±0.8 | 4.3±0.9 | 28.0±1.5 | +0.7 |
| | **FedNCN** | **55.0±1.8** | **13.3±2.5** | **19.7±3.7** | **24.8±1.4** | **+9.2** | | **FedNCN** | **57.5±1.6** | **19.7±1.9** | **18.0±1.7** | **32.8±1.1** | **+9.9** |
| PubMed (5 Clients) | Local | 55.5±2.1 | 9.5±1.3 | 9.5±1.5 | 38.1±0.7 | - | PubMed (20 Clients) | Local | 54.3±0.9 | 6.3±0.8 | 4.9±1.2 | 35.4±1.0 | - |
| | FedAvg | 61.7±0.5 | 9.6±0.4 | 9.0±0.9 | 30.8±1.8 | -0.4 | | FedAvg | 64.2±3.2 | 4.6±2.3 | 5.1±1.5 | 20.0±3.8 | -1.8 |
| | FedProx | 60.5±1.7 | 10.3±4.2 | 11.0±3.6 | 39.8±2.0 | +2.3 | | FedProx | 60.8±1.4 | 6.1±1.3 | 7.4±1.3 | 36.7±0.8 | +2.5 |
| | FedPer | 56.1±1.7 | 8.7±0.7 | 10.6±2.0 | 38.7±2.3 | +0.4 | | FedPer | 53.3±0.4 | 6.2±0.8 | 4.5±0.8 | 35.2±0.5 | -0.4 |
| | **FedNCN** | **63.4±1.6** | **10.5±3.5** | **13.9±3.6** | **44.0±2.3** | **+4.8** | | **FedNCN** | **66.3±1.7** | **7.5±0.9** | **11.0±1.2** | **41.3±1.3** | **+6.3** |
| Amazon-Computer (5 Clients) | Local | 37.7±1.7 | 2.9±0.2 | 1.0±1.3 | 13.2±0.7 | - | Amazon-Computer (20 Clients) | Local | 48.1±0.5 | 3.3±0.1 | 0.0±0.5 | 18.2±0.3 | - |
| | FedAvg | 59.6±0.3 | 0.9±0.2 | 0.1±0.1 | 11.4±0.4 | -4.3 | | FedAvg | 64.2±2.0 | 1.7±0.2 | -0.4±0.1 | 17.9±0.2 | +3.5 |
| | FedProx | 39.2±3.9 | 2.2±0.3 | 0.5±1.4 | 13.1±0.7 | -0.1 | | FedProx | 50.2±1.6 | 3.1±0.2 | 1.7±0.4 | 19.9±0.4 | +1.3 |
| | FedPer | 45.7±1.8 | 2.1±0.2 | 0.7±0.8 | 13.3±1.0 | +1.8 | | FedPer | 52.9±1.5 | 3.6±0.1 | 2.4±0.8 | 20.2±0.6 | +2.4 |
| | **FedNCN** | **63.4±2.3** | **17.0±9.0** | **15.3±8.7** | **21.4±5.9** | **+15.6** | | **FedNCN** | **70.2±2.6** | **21.6±1.5** | **22.9±3.2** | **32.5±0.8** | **+19.4** |
| Amazon-Photo (5 Clients) | Local | 47.0±0.6 | 4.7±0.3 | 3.8±0.9 | 16.3±0.5 | - | Amazon-Photo (20 Clients) | Local | 55.1±0.9 | 9.7±0.2 | 8.6±0.3 | 25.5±0.5 | - |
| | FedAvg | 63.4±0.5 | 0.9±0.3 | 0.0±0.3 | 12.3±0.6 | +1.2 | | FedAvg | 65.5±2.2 | 6.4±2.9 | 5.0±2.5 | 23.2±4.9 | +0.3 |
| | FedProx | 49.1±1.2 | 5.2±1.0 | 7.7±1.4 | 17.1±0.7 | +1.8 | | FedProx | 58.0±2.1 | 10.2±0.4 | 8.4±0.7 | 26.7±0.6 | +1.1 |
| | FedPer | 55.2±1.6 | 5.5±1.1 | 7.1±1.6 | 16.4±0.9 | +3.1 | | FedPer | 63.4±0.8 | 12.2±0.9 | 11.7±1.3 | 27.9±0.8 | +4.1 |
| | **FedNCN** | **74.1±3.0** | **37.2±5.3** | **43.1±7.0** | **29.1±2.8** | **+27.9** | | **FedNCN** | **66.1±2.1** | **19.9±1.6** | **16.7±2.0** | **32.5±0.9** | **+9.1** |
| Questions (5 Clients) | Local | 75.1±2.4 | 0.4±0.3 | -0.7±0.5 | 44.7±1.0 | - | Questions (20 Clients) | Local | 80.7±1.6 | 0.7±0.2 | -0.4±0.5 | 46.5±0.4 | - |
| | FedAvg | 87.4±0.8 | 0.0±0.0 | 0.0±0.0 | 46.2±0.0 | +3.5 | | FedAvg | 88.2±1.5 | 0.0±0.0 | -0.1±0.0 | 46.2±0.0 | +1.7 |
| | FedProx | 85.3±4.1 | 0.3±0.1 | 1.6±1.1 | 48.8±1.5 | +4.1 | | FedProx | 81.7±0.8 | 0.6±0.2 | -0.1±1.2 | 47.0±0.5 | +0.4 |
| | FedPer | 84.1±2.0 | 0.3±0.1 | 1.1±0.9 | 48.4±0.8 | +3.6 | | FedPer | 81.3±0.5 | 0.5±0.2 | -0.2±0.8 | 46.9±0.6 | +0.3 |
| | **FedNCN** | **94.6±2.0** | **1.9±0.9** | **6.4±2.9** | **52.9±1.7** | **+9.1** | | **FedNCN** | **93.0±5.1** | **1.1±0.8** | **2.9±3.3** | **50.8±2.3** | **+5.1** |

## D.2. Experiment Results

**Comparison with Supervised FGL Methods** In the main text, we compare the proposed FedNCN with FedSage+, FedPUB, and FedTAD, demonstrating the superiority of our method. Here, we compare FedNCN with FedGTA and FedIIH to further validate the competitiveness of FedNCN in clustering tasks. The experimental setup remains consistent with that in the main text, where each supervised federated node-level classification method is trained using 5% of labeled graph data from the dataset. As shown in Fig. 7, the experimental results indicate that, under different client partitions, FedNCN consistently outperforms federated node-level classification tasks (i.e., FedGTA and FedIIH) with limited labeled data.

**Comparison of FL Aggregation Strategies** In the main text, we demonstrate that the proposed FedNCN method significantly outperforms three classical FL aggregation strategies (including FedAvg, FedProx, and FedPer) in 10 clients. Here, as shown in Table 5, we report the performance of FedNCN in 5 clients and 20 clients across multiple clustering metrics on five unlabeled benchmark graph datasets. As seen, we can observe these findings: 1) in the 5 clients, FedNCN achieves average improvements of 9.2%, 4.8%, 15.6%, 27.9%, and 9.1% on the CiteSeer, PubMed, Amazon-Computer, Amazon-Photo, and Question datasets, respectively; 2) in the 20 clients, the average improvements are 9.9%, 6.3%, 19.4%, 9.1%, and 5.1% on the same datasets, respectively. These findings indicate that under different client partitions, our method significantly enhances the clustering performance on all datasets compared to other aggregation strategies with our local model. These improvements could be attributed to the following points. Firstly, the cross-subgraph link mending strategy restores links as much as possible, facilitating better information communication. This allows the model to learn consistent prototypes and model parameters from the mended graph, thereby eliminating the constraints caused by the destroyed structural information and enabling each local model to generate a clustering-friendly target distribution.

**Graph Kernel Ablation Analysis** In this part, we evaluate the effectiveness of different graph kernels in FedNCN. In our setup, "COS" denotes the integration of the COS graph kernel with FedNCN, while "WL," "SP," and "LT" denote the combinations of the WL, SP, and LT graph kernels with FedNCN, respectively. As shown in Fig. 8, we observe that 1) the ACC remains consistently above 50% across all experiments on different datasets; 2) under different datasets and client partitions, the choice of graph kernels has little impact on clustering performance. These findings may be attributed to the fact that different graph kernels focus on different paradigms and the inherent patterns of graph structures. However, in our method, which benefits from the effective recovery of intra-cluster links, these graph kernels can always identify representative features of refined subgraph structures to accurately measure the affinity between these subgraphs. In summary, each graph kernel method has its own strengths, but the overall performance differences are minimal. Therefore,

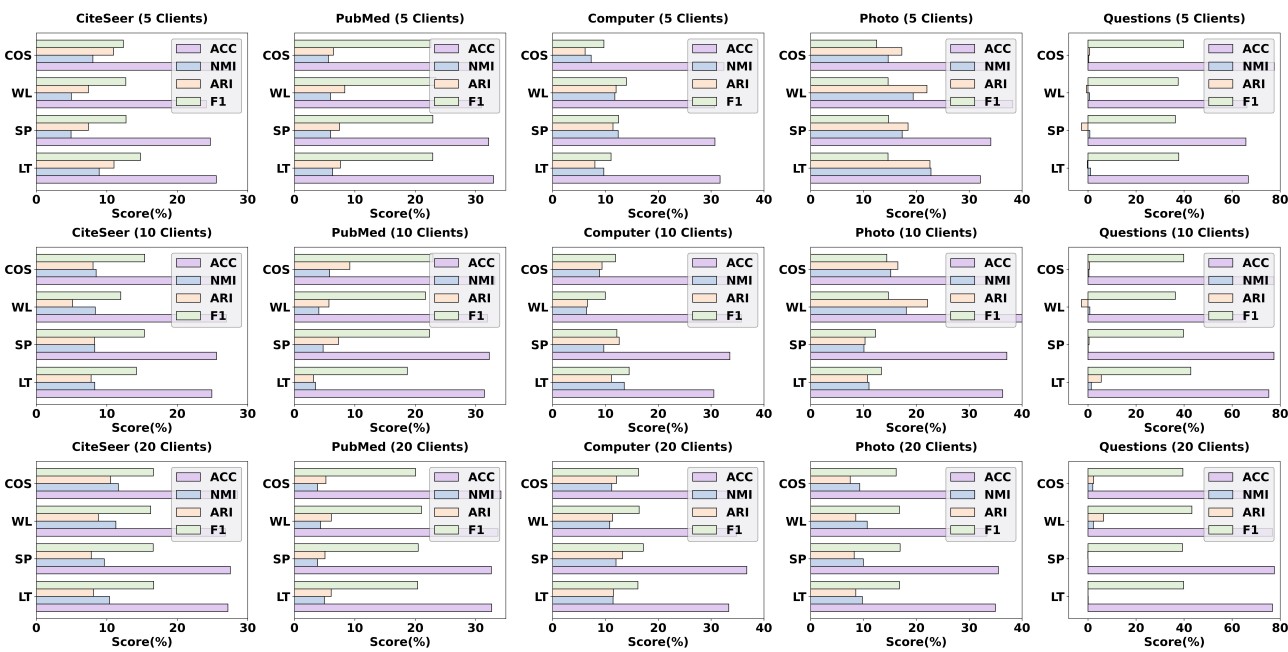

*Figure 8.* The performance of different graph kernel ablation studies on five benchmark datasets (CiteSeer, PubMed, Computer, Photo, Questions). Here, each row represents different client settings (5 clients, 10 clients, 20 clients).

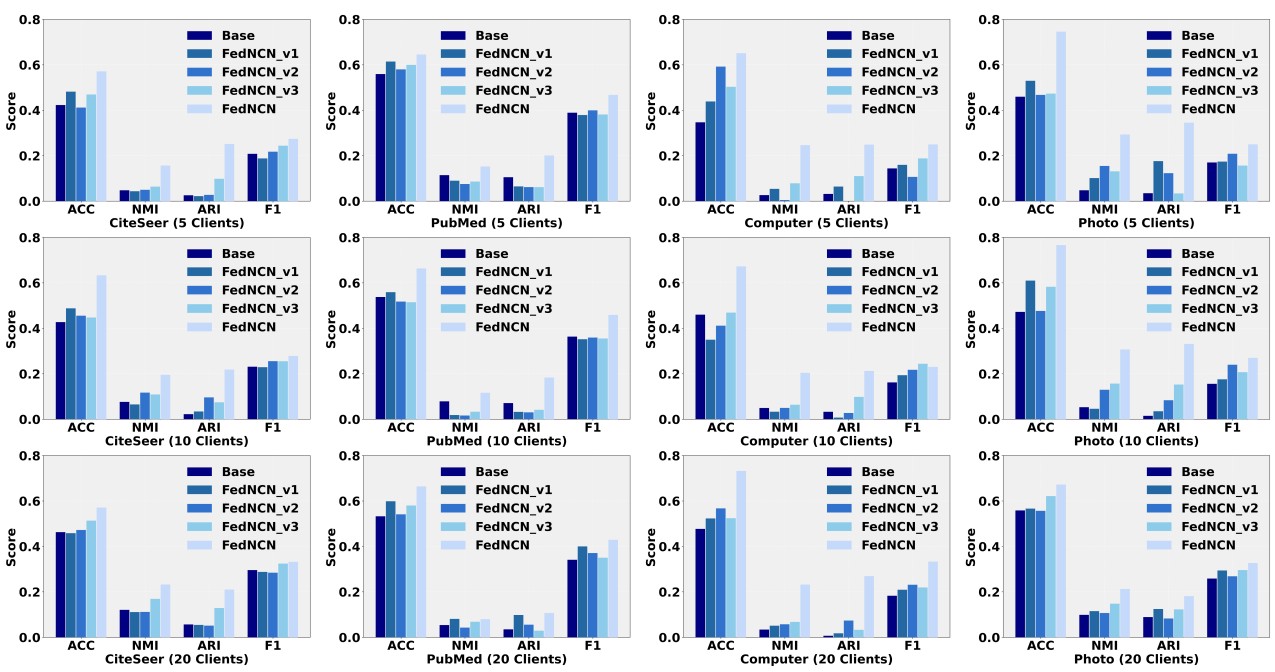

*Figure 9.* The performance of module ablation in FedNCN under the federated node-level clustering scenario.

in our approach, we choose the computationally simpler "COS" graph kernel for ease of extension and application.

**Module Ablation Study** In this section, we evaluate the effectiveness of the proposed modules in FedNCN. In our setup, "Base" denotes the local model of FedNCN. "FedNCN_v1", "FedNCN_v2", and "FedNCN_v3" denote three FedNCN variants with the intra-cluster link mending strategy, the inter-cluster link mending strategy, and the improved N-Cut method being removed, respectively. As summarized in Fig. 9, we can observe that 1) the clustering metrics of the FedNCN method are consistently better than the other methods; 2) Compared to the "Base", "FedNCN_v1", "FedNCN_v2", and "FedNCN_v3" exhibit better clustering performance in most cases. These findings can be attributed to the following points. Firstly, FedNCN can effectively restore the missing links between clients, facilitating better knowledge sharing to enhance each local model for better clustering. Secondly, each module of FedNCN plays an effective role in restoring the missing links between clients on the server.

