# OpenReview forum: "Federated Node-Level Clustering Network with Cross-Subgraph Link Mending"
_ICML.cc/2025/Conference — ICML 2025 poster_

### Official Review · Reviewer_ZDxr · 2025-03-07

**Overall Recommendation:** 4

**Summary:**

In this study, the authors investigate two unexplored issues in federated graph learning (FGL), namely: 1) the heavy reliance on labeled graph samples that are difficult to obtain in real-world applications; and 2) the inevitable missing links caused by partitioning a complete graph into several subgraphs. To address these issues, the authors propose an easy-to-understand federated learning algorithm, named FedNCN, which introduces a dynamic graph construction scheme to mend the missing links among subgraphs. The main results on benchmark datasets show improvements in clustering performance compared to existing methods.

**Claims And Evidence:**

The evidence provided in support of the claims in this paper is convincing.

**Essential References Not Discussed:**

No, the related works that are essential to understanding the key contributions of the paper have been included in the manuscript.

**Experimental Designs Or Analyses:**

Through a series of experiments, the authors verify the effectiveness of the method. Compared to existing advanced FGL methods, FedNCN achieves better results.

**Methods And Evaluation Criteria:**

Yes, the evaluation criteria make sense.

**Other Comments Or Suggestions:**

No further comments or suggestions.

**Other Strengths And Weaknesses:**

Strengths:
The article is well-motivated. It is meaningful and reasonable to provide some insights about the newly designed choices, which are valuable for researchers in related areas. In addition, some experimental results seem good.

Weaknesses:
(W1) There are still some issues that should be further discussed. For example, in Table 1 on page 6, although the proposed method achieves the best performance on the Questions dataset compared to the advanced FGL method, its NMI and ARI scores are relatively low compared to other datasets. What are the potential reasons for this phenomenon? FedNCN first maximizes edge construction and then minimizes edge retention. In future work, adversarial learning [1, 2] could be used to further refine the entire process of FedNCN.
[1] Gong L, Zhou S, Tu W, et al. Attributed Graph Clustering with Dual Redundancy Reduction[C]//IJCAI. 2022: 3015-3021.
[2] Suresh S, Li P, Hao C, et al. Adversarial graph augmentation to improve graph contrastive learning[J]. Advances in Neural Information Processing Systems, 2021, 34: 15920-15933.

(W2) The text contains some redundant expressions that should be optimized. For example: 1) Left column, lines 331-334: "Here, 'Local' denotes the use of only our local model, while in 'FedAvg', 'FedProx', and 'FedPer', different aggregation methods are applied by the server, with the client all using our local model." This content has already been described in the main text and is reiterated in the table caption, leading to redundancy. It is recommended to express it only once to avoid repetition. 2) Right column, lines 344-346: "'BS' denotes the local model of FedNCN. 'BS+GKS' denotes the FedNCN without mending the missing links, and 'BS+GKS+CLM' denotes the FedNCN." This description is repetitive and can be condensed to improve readability and logical clarity.

**Questions For Authors:**

No

**Relation To Broader Scientific Literature:**

This topic and the obtained findings are interesting to the federated graph learning community

**Theoretical Claims:**

The paper primarily focuses on experimental validation, and no major theoretical inconsistencies were identified.

---

> ### Author Rebuttal · Authors · 2025-04-01
>
> # Response to Reviewer:
>
> **(W1) :**
>
> **Reasons for the low NMI and ARI:** Thanks. The Questions dataset exhibits a significant class imbalance, with 47461 samples for class 0 and 1460 for class 1, which could result in lower NMI and ARI performance. Although class imbalance presents significant difficulties, our approach demonstrates competitive results relative to existing advanced methods. In future work, we will further improve FedNCN to enhance its performance in scenarios with class imbalance.
>
> **Future optimization directions:** Thank you for your valuable suggestion. We agree that our method could indeed be further improved by incorporating adversarial learning [1, 2] in future work. Specifically, there are two steps in our paper about maximizing edge construction and then minimizing edge retention. Transforming these current two steps into a dynamic optimization framework using adversarial learning could enhance the robustness of FedNCN in server mending missing links. We will consider this direction in our future work to further refine the proposed FedNCN.
>
> **(W2) :**
>
> **Expression:** Thanks. We have tried our best to avoid redundant expressions according to your advice. In addition, we will further refine the writing in the final version of the paper to make it more concise.
>
> Thanks for your constructive comments, we hope that our responses will make you satisfied.

---

> > ### Comment · Reviewer_ZDxr · 2025-04-04
> >
> > Thanks for the authors' response, most of my previous concerns have been addressed. As discussed in the response, introducing the idea of adversarial learning has the potential to further improve the current version. Additionly, I would like to know how the performance of FedNCN would be if we directly upload the representative signals to construct the global graph without maximizing edge construction and minimizing edge retention.

---

> > > ### Author Response · Authors · 2025-04-06
> > >
> > > # Response to Reviewer:
> > >
> > > Thank you for your follow-up and constructive feedback. To address your question, we have conducted additional experiments to compare the clustering performance between the proposed FedNCN and its variants. Specifically, in our setups, the "FedNCN_v1" method denotes a variant of FedNCN that directly uploads representative signals to construct a global graph for global missing link recovery. Moreover, the "Local" method denotes the local model of the proposed FedNCN. For convenience, we present the corresponding results in the hyperlink due to the space limit. As seen in this URL: https://anonymous.4open.science/r/FedNCN-rebu2-FF6C/metric.png, several major observations can be found: 1) compared to the "Local" method and the "FedNCN_v1" method, FedNCN produces ACC performance gains of 26.07% and 28.65% on the Photo dataset in the non-overlapping setting with 5 clients, indicating that the proposed cross-subgraph links mending strategy plays an essential role in effectively handling federated node-level clustering; 2) compared to the "Local" method, the "FedNCN_v1" method even performs worse in many cases. For instance, on the Computer dataset in non-overlapping setting with 10 clients, the ACC of the "FedNCN_v1" method drops by 10.03% compared to the "Local" method. These findings demonstrate that the direct application of KNN graph construction for global link recovery inadequately captures the relationships between intra-cluster and inter-cluster nodes in the uploaded signals, resulting in sub-optimal performance.
> > >
> > > We hope that our responses will address your concerns.

---

### Official Review · Reviewer_S8JV · 2025-03-11

**Overall Recommendation:** 4

**Summary:**

This paper designs a federated graph learning framework called Federated Node-Level Clustering Network (FedNCN) that mends the cross-subgraph missing links to enhance the clustering performance of each client in an unlabeled circumstance while not sharing private data. The work also conducts experiments on several benchmark datasets using the proposed method.

**Claims And Evidence:**

Yes.

**Essential References Not Discussed:**

No.

**Experimental Designs Or Analyses:**

Yes, the author has provided a detailed analysis.

**Methods And Evaluation Criteria:**

Yes, I believe that the methods and evaluation criteria presented in this paper are meaningful for the proposed problem and have been clearly described.

**Other Comments Or Suggestions:**

See the weaknesses.

**Other Strengths And Weaknesses:**

Pros:
- In contrast to existing unsupervised FGL methods, FedNCN is proposed to address the issue of missing links caused by graph partition, which reveals a certain novelty.
- The motivations are presented clearly, and each innovation has an intuitive explanation. In particular, the proposed cross-subgraph link mending strategy is a significant technical contribution to the field of node-level FGL.
- The results show that FedNCN consistently outperforms existing methods, even when fewer real labels are used. This robust experimental evidence supports the paper's claims.

Cons:
- Both the local model and the global model in FedNCN compute cluster centers. What is the difference between these two types of clustering centers? Moreover, are these two cluster centers updated iteratively, or are they calculated only once? The authors need to give an instruction to make the reader understand them.
- In Table 1, what does the "*" next to the methods signify? The paper does not provide an explanation for this. Could the author provide a clear explanation for the readers?
- To enhance the reproducibility of the experiments, the authors should share the source code of the proposed method.
- The limitations are not discussed in the paper. The authors need to provide a brief discussion on the limitations of the study to offer a more balanced and comprehensive evaluation of FedNCN. This would provide valuable guidance for future research in unsupervised FGL.
- Some types need to be carefully checked and corrected, such as, “... crucial role in FedGCN...” --> “... crucial role in FedNCN...”.

**Questions For Authors:**

See the weaknesses.

**Relation To Broader Scientific Literature:**

Different from existing supervised federated graph learning methods, the proposed federated learning framework is the first to explore the issue of missing links in an unsupervised setting.

**Theoretical Claims:**

The stability guarantee provided by FedNCN is both interesting and well-supported.

---

> ### Author Rebuttal · Authors · 2025-04-01
>
> # Response to Reviewer:
>
> **1. Clustering centers:** Thanks. In FedNCN, both the client and the server have models that are used to learn the graph embeddings. The method for calculating the cluster centers is the same for both, initialized by K-means and updated iteratively by the model. However, the former (local model) learns the local graph data assigned to each client to obtain key clustering signals for subsequent cross-subgraph links restoration, while the latter (global model) learns the mended graph to obtain consensus prototypes that guide each local model to better cluster.
>
> **2. Explanation:** Thanks. In our manuscript, the methods marked with \* indicate that supervised methods are adapted to an unsupervised scenario.
>
> **3. Source code:** Thanks. The complete source code and datasets will be available for reproducibility if the paper is accepted.
>
> **4. Limitations and future work:** Thanks. Our method is designed for federated node-level tasks with cross-subgraph link mending in the unsupervised scenario. However, many scenarios involve federated graph-level clustering in practical applications, such as social network analysis and disease prediction. In federated graph-level clustering, each client learns from independent graph-level datasets, which exhibit significant structural heterogeneity. The divergence in multi-source data sharing is further exacerbated due to the lack of label guidance, making it difficult for our current model to effectively address these issues. Future work can explore a cross-domain federated learning framework to be applied to broader scenarios.
>
> **5. Detailed issues:** Thanks for your careful review. We have revised the sentence “the hyperparameter $k$ indeed plays a crucial role in FedGCN, ...” to “the hyperparameter $k$ indeed plays a crucial role in FedNCN, ...”. We will try our best to double-check the manuscript carefully and correct similar typos in our final version.
>
> Thanks for your constructive comments, we hope that our responses will make you satisfied.

---

### Official Review · Reviewer_c9LU · 2025-03-12

**Overall Recommendation:** 4

**Summary:**

The authors propose a Federated Node-Level Clustering Network (FedNCN), which is the first attempt to tackle the issue of link missing caused by graph partition in an unsupervised learning scenario. The core idea of FedNCN is to mend the destroyed links using prior clustering knowledge. Extensive experiments have been conducted to evaluate the performance of FedNCN.

**Claims And Evidence:**

Yes, the claims made in the submitted paper are supported by clear and convincing evidence.

**Essential References Not Discussed:**

All the references that are crucial to the key contributions of this paper have been cited and discussed.

**Experimental Designs Or Analyses:**

The experimental setup is clearly presented, and the comparative experiments, ablation studies, and hyperparameter analysis are comprehensively covered.

**Methods And Evaluation Criteria:**

Yes, this paper uses benchmark datasets and widely accepted evaluation metrics in the field.

**Other Comments Or Suggestions:**

All concerns are presented as weaknesses.

**Other Strengths And Weaknesses:**

Strengths
1. Overall, I think this is a good work that takes a step forward in unsupervised federated node-level graph learning for missing links. This work effectively leverages the prior learned clustering knowledge to enable the model to accurately conduct cross-subgraph link mending, which promotes the great clustering encoding capacity of the local model.
2. The problem addressed in this paper is evident, and the innovations are novel, demonstrating its research value. Moreover, the overall structure of the paper is well-organized. Necessary illustrative figures are provided to help readers understand the contents.
3. Experiments on five graph benchmark datasets demonstrated the effectiveness and superiority of the proposed FedNCN against its competitors. Furthermore, ablation studies and convergence analysis further confirm its strong potential for practical applications.


Weaknesses
1. The authors should further explain the relationship among the three proposed components, i.e., the local model learning strategy, the cross-subgraph link mending strategy, and global knowledge sharing strategy.
2. A few errors need to be checked and corrected. For example, in Figure 2, do 'clustering signals' and 'uploaded signals' refer to the same entities? If so, they should be consistently noted in the figure.
3. I have some questions about certain contents in this paper. In the left column on page 6, lines 320-322: “Here, we consider the scenario where there are no overlapping nodes between subgraphs. ” What does the scenario with overlapping nodes refer to, and why is this scenario not considered in this paper? Are there any other possible scenarios?

**Questions For Authors:**

Please check the weaknesses in “Other Strengths And Weaknesses” section.

**Relation To Broader Scientific Literature:**

The paper proposes a federated graph learning framework that is capable of mending the destroyed subgraph links across clients.

**Theoretical Claims:**

The author suppose that connected nodes typically have high feature similarity, which is evident and has been used in previous work.

---

> ### Author Rebuttal · Authors · 2025-04-01
>
> # Response to Reviewer:
>
> **1. Relationships between three components:** Thanks. In our approach, the local model learning strategy collects and preserves more reliable clustering signals to prepare for the recovery of damaged sample connections. The cross-subgraph link mending strategy utilizes the prior learned clustering knowledge to establish correct links between subgraphs, which provides high-quality data for global consensus learning. The global knowledge sharing strategy learns the clustering-friendly consensus prototypes based on the mended graph and ensures reliable feedback for each client, enhancing the discriminative ability of each local model in graph-level embeddings. These three components are seamlessly integrated into a unified optimization framework.
>
> **2. Fig. 2 revision:** Thanks for your careful review. In Figure 2, 'clustering signals' and 'uploaded signals' refer to the same entities, and we have revised the notation for consistency according to your advice. We will update Fig. 2 in the final version.
>
> **3. Definition of “overlapping”:** The term "overlapping" has been widely present in previous studies on federated graph learning [1, 2], referring to the scenario where different subgraphs are distributed across multiple clients and share some nodes. In contrast, since the problem definition in our paper involves the structure loss caused by partitioning a complete graph into several subgraphs, we do not consider node overlapping and instead focus on the scenario where there are no overlapping nodes between subgraphs.
>
> [1]Baek, J.; Jeong, W.; Jin, J.; Yoon, J.; and Hwang, S. 2023. Personalized subgraph federated learning. In *ICML*, 1396-1415.
>
> [2] Zhu, Y.; Li, X.; Wu, Z.; Wu, D.; M, Hu.; and Li, R. 2024. FedTAD: Topology-aware Data-free Knowledge Distillation for Subgraph Federated Learning. In *IJCAI*, 5716–5724.
>
> Thanks for your constructive comments, we hope that our responses will make you satisfied.

---

### Official Review · Reviewer_uoNs · 2025-03-13

**Overall Recommendation:** 3

**Summary:**

This paper introduces federated node-level clustering that achieves cross-subgraph link mending under unsupervised circumstances. The proposed approach is mainly composed of three components, i.e., the local model learning scheme that collects and preserves trustworthy clustering signals for destroyed sample link restoration, the cross-subgraph links mending scheme that establishes correct links among subgraphs with the aid of prior learned clustering knowledge, and the global knowledge sharing scheme that learns high-quality consensus features based on the mended graph and ensures reliable feedback to each client. Abundant experiments on five benchmark datasets have been done.

**Claims And Evidence:**

The authors claim that they use graph kernel similarity and N-cut to dynamically construct the mended graph
However, there seems to be a lack of discussion about the intuition behind using such a method. For example, why use dynamic graph construction instead of KNN graph construction?

**Essential References Not Discussed:**

This paper utilizes federated learning to train GCN models with reduced communication overhead, and the work on distributed graph learning with cross-client edges using homomorphic encryption [1] should be discussed.

[1] Yao, Yuhang, et al. "FedGCN: convergence-communication trade-offs in federated training of graph convolutional networks." In NeurIPS 2023.

**Experimental Designs Or Analyses:**

Yes, the experimental results are sufficient, and the corresponding analyses are self-consistent.

**Methods And Evaluation Criteria:**

The authors employ five benchmark datasets (i.e., CiteSeer, PubMed, Amazon-Computer, Amazon-Photo, and Questions) and four widely used evaluation metrics (i.e., ACC, NMI, ARI, and F1) to evaluate the proposed method.

**Other Comments Or Suggestions:**

N/A.

**Other Strengths And Weaknesses:**

Advantages:

i. Novelty and Innovation: This paper proposes a new federated node-level clustering network for cross-subgraph link restoration, which is a fresh perspective in federated graph learning.

ii. Technical Contribution: The authors design three components (the local model learning, the cross-subgraph links mending, and the global knowledge sharing) that are seamlessly integrated into a unified optimization framework, offering a more  reasonable way to mend cross-subgraph links.

Disadvantages:

i. Some concerns: a) Can you describe what is the clustering ground truth in this task? Is it simply the node label? b) How are the parameters of each local model initialized in the federated learning framework? How is the initialization of the learnable weight matrix W achieved? More explanations should be given. c) Federated graph-level clustering also represents a variant of unsupervised federated graph learning. Could the proposed network be extended to handle this task? d) It would be better that the authors further prove the advantages of dynamic graph construction compared to the traditional KNN-based approach.

ii. Minor writing issues: a) The dataset names are not consistent. For example, Table 1 and Table 2 use 'Amazon-Computer' and 'Amazon-Photo,' while Figures 3, 4, 5, and 6 use 'Computer' and 'Photo.' While these do not significantly affect the content, they should be corrected to improve professionalism and readability. b) The presence of unnecessary punctuation marks (e.g., extra period at line 104 in the left column).

**Questions For Authors:**

a) Can you describe what is the clustering ground truth in this task? Is it simply the node label?

b) How are the parameters of each local model initialized in the federated learning framework? How is the initialization of the learnable weight matrix W achieved? More explanations should be given.

c) Federated graph-level clustering also represents a variant of unsupervised federated graph learning. Could the proposed network be extended to handle this task?

d) It would be better that the authors further prove the advantages of dynamic graph construction compared to the traditional KNN-based approach.

**Relation To Broader Scientific Literature:**

The paper builds on recent advances in federated graph learning models, extending ideas from prior literature such as FedPUB and FedTAD. It attempts to introduce a novel method of dynamic graph construction in unsupervised settings, using graph kernel similarity and N-cut to enhance the clustering performance of each local model.

**Theoretical Claims:**

Yes. I have checked. It would be better that the authors further prove the advantages of dynamic graph construction compared to the traditional KNN-based approach.

---

> ### Author Rebuttal · Authors · 2025-04-01
>
> # Response to Reviewer:
>
> **i. Some concerns:**
>
> **a) Clustering ground truth:** Thanks for your question. The clustering ground truth is the true cluster category of samples. In our paper, it corresponds to the node label. The ground truth is used only in the evaluation stage of this task.
>
> **b) Parameter initialization:** Thanks. We do the initialization by following the parameter initialization principles in federated learning referring to previous work [1, 2]. The learnable weight matrix **W** is randomly initialized at the global server and subsequently delivered to the local models for learning.
>
> [1] Baek, J.; Jeong, W.; Jin, J.; Yoon, J.; and Hwang, S. 2023. Personalized subgraph federated learning. In *ICML*, 1396-1415.
>
> [2] Zhu, Y.; Li, X.; Wu, Z.; Wu, D.; M, Hu.; and Li, R. 2024. FedTAD: Topology-aware Data-free Knowledge Distillation for Subgraph Federated Learning. In *IJCAI*, 5716–5724.
>
> **c) Federated graph-level clustering:** Thanks. It is indeed an interesting and meaningful task. Federated node-level clustering clusters nodes in only one graph, while federated graph-level clustering clusters multiple graphs. Both clustering methods work in the way of federated learning. The proposed network encounters several difficulties when handling the federated graph-level clustering task. For instance, compared to node-level tasks, graph-level tasks involve data that may originate from different domains and have more complex graph structures. This makes it difficult to capture common patterns across multiple clients. In future work, we aim to overcome these challenges and extend our model to federated graph-level clustering tasks.
>
> **d) Advantages:** Thanks. Due to the non-uniform distribution of instances in the sample space, the traditional KNN graph construction method has inherent flaws [1-3]. Using different values of $K$ for different classes outperforms using a fixed $K$ across all classes, Li et al. have drawn this conclusion through mathematical derivation. Based on this theoretical claim, we further demonstrate the advantages of dynamic graph construction through experiments, compared to the traditional KNN-based approach. The experimental results have been provided at the anonymous link https://anonymous.4open.science/r/FedNCN-rebu-418F/data.png
>
> [1] S, Zhang. 2020. Challenges in KNN Classification. *IEEE Transactions on Knowledge and Data Engineering*, 4663-4675.
>
> [2] S, Kazemi.; R, Goel.; K, Jain.; I, Kobyzev.; A, Sethi.; P, Forsyth.; and P, Poupart. 2020. Representation learning for dynamic graphs: A survey. *Journal of Machine Learning Research*, 1-73.
>
> [3] M, Munir.; W, Avery.; M, Rahman.; and R, Marculescu.; 2024. Greedyvig: Dynamic axial graph construction for efficient vision gnns. In *CVPR*, 6118-6127.
>
> [4] B. Li.; Y. Chen.; and Y. Chen. 2008. The nearest neighbor algorithm of local probability centers", In *IEEE Transactions on Systems, Man, and Cybernetics: Systems*. 141-154.
>
> **ii. Minor writing issues:**
>
> **Typos:** Thanks for your careful review again! These typos have been revised. a) We have made the dataset names consistent in Table 1, Table 2, and Figures 3-5. b) We have removed unnecessary symbols (e.g., the extra period at line 104 in the left column). Moreover, we have tried our best to correct similar typos and double-checked throughout the paper.
>
> **Essential References Not Discussed:**
>
> **Adding the reference:** Thanks for your careful review again! The FedGCN [1] method that you mentioned has been discussed in the final version.
>
> [1] Yao, Yuhang, et al. "FedGCN: convergence-communication trade-offs in federated training of graph convolutional networks." In NeurIPS 2023.
>
> Thanks for your constructive comments, we hope that our responses will make you satisfied.

---

> > ### Comment · Reviewer_uoNs · 2025-04-06
> >
> > Thank you for the author's response. I have also reviewed the comments from the other reviewers and the corresponding replies from the author. I will maintain my score.

---

> > > ### Author Response · Authors · 2025-04-07
> > >
> > > Thank you for your support and thoughtful suggestions. We will carefully revise the final version based on your valuable feedback.

---

### Decision · Program_Chairs · 2025-05-01

**Decision:**

Accept (poster)

**Comment:**

This paper received four effective reviews, and all of them are positive. Overall, the paper is of good quality and should be accepted.